# Joint inference of exclusivity patterns and recurrent trajectories from tumor mutation trees

Xiang Ge Luo [1,2], Jack Kuipers [1,2] & Niko Beerenwinkel [1,2] ✉

Cancer progression is an evolutionary process shaped by both deterministic and stochastic forces. Multi-region and single-cell sequencing of tumors enable high-resolution reconstruction of the mutational history of each tumor and highlight the extensive diversity across tumors and patients. Resolving the interactions among mutations and recovering recurrent evolutionary processes may offer greater opportunities for successful therapeutic strategies. To this end, we present a novel probabilistic framework, called TreeMHN, for the joint inference of exclusivity patterns and recurrent trajectories from a cohort of intra-tumor phylogenetic trees. Through simulations, we show that TreeMHN outperforms existing alternatives that can only focus on one aspect of the task. By analyzing datasets of blood, lung, and breast cancers, we find the most likely evolutionary trajectories and mutational patterns, consistent with and enriching our current understanding of tumorigenesis. Moreover, TreeMHN facilitates the prediction of tumor evolution and provides probabilistic measures on the next mutational events given a tumor tree, a prerequisite for evolution-guided treatment strategies.

Tumors emerge and develop malignancy through a somatic evolutionary process of accumulating selectively advantageous mutations in cells[1]. The genetic and phenotypic clonal diversity within a tumor, also known as intratumor heterogeneity (ITH), enables tumor cells to quickly adapt to micro-environmental changes, including those induced by treatment, often leading to lethal outcomes due to metastases or drug resistance[2,3]. Despite the inherent stochasticity of tumor evolution, recent evidence supported by increasingly available data and new computational methods has revealed at least some repeated features in tumor progression, such as frequently mutated genes[4], specific order constraints on mutations[5,6], and repeated evolutionary trajectories[7–10]. The ability to recover reproducible features of cancer evolution, and more importantly, to make reliable predictions of future evolutionary steps, is crucial for the development of successful therapeutic interventions[11–14].

Recent advances in multi-region sequencing[15,16], single-cell sequencing[17], and phylogenetic tree inference[18,19] enable more precise characterization of clonal architecture and provide a clearer picture of tumor evolution. However, the increased resolution further substantiates the extensive variability in the subclonal compositions and mutational histories between tumors, making it more challenging to infer repeatable elements. For example, two parallel studies of acute myeloid leukemia (AML) using single-cell panel sequencing[20,21] show that the reconstructed trees typically contained a small number of clones based on the specific driver mutations that were part of the panel and that they varied considerably between any two patients. In particular, both studies report over-represented pairs of co-occurring or clonally exclusive mutations in the subclones. These relationships have recently been examined with a customized statistical test, called GeneAccord[22], which is designed for evolutionary-related samples from the same tumor. Mutations that co-occur more frequently in the same clonal lineage may indicate synergistic effects on cell proliferation and survival[4]. Clonally exclusive mutations, on the other hand, occur more frequently in different lineages. They may suggest either clonal

¹Department of Biosystems Science and Engineering, ETH Zurich, Mattenstrasse 26, 4058 Basel, Switzerland. ²SIB Swiss Institute of Bioinformatics, Mattenstrasse 26, 4058 Basel, Switzerland. ✉e-mail: niko.beerenwinkel@bsse.ethz.ch

cooperation, where cell populations harboring complementary sets of mutations collaborate to promote tumor growth[23], or synthetic lethality, meaning that acquiring both mutations on the same genome will significantly reduce the viability of the clone[24]. Since clonal interactions have a major impact on intratumor heterogeneity and the observed evolutionary trajectories, it would be natural to incorporate mutational interdependencies within and between subclones when modeling tumor progression.

For cross-sectional bulk sequencing data, where each tumor is summarized by a binary genotype, a collection of probabilistic methods for inferring the temporal order of mutations and predicting tumor progression is called cancer progression models (CPMs)[25,26]. In light of the observation that mutually exclusive mutations are often associated with genes in the same functional pathway, Raphael & Vandin developed the first CPM that jointly infers sets of mutually exclusive mutations and the temporal order among them using a chain model[27]. Later, pathTiMEx[28] generalized the linear structure with a continuous-time Conjunctive Bayesian Network (CBN)[29]. A recent method, called Mutual Hazard Networks (MHNs), does not explicitly group exclusive mutations but re-parameterizes mutational waiting times using a matrix that encodes both co-occurrence and exclusivity[30]. However, these methods require independent samples of binary genotypes as input. As such, they have not been designed for tree-structured data as they cannot capture the subclonal structure and dependencies within a tumor nor utilize the existing order information from the tumor phylogenies. In addition, mutual exclusivity is defined at the patient level, whereas clonal exclusivity refers to pairs of mutations that occur less frequently in the same subclone but can still co-exist in the same tumor[22]. Hence, the consensus genotype of a tumor can contain some or all of the clonally exclusive mutations, but the above CPMs will treat them as evidence of co-occurrence.

First efforts have been made to infer recurrent evolutionary trajectories in a cohort of tumor phylogenies reconstructed from bulk, multi-region, or single-cell sequencing data. Based on transfer learning, REVOLVER[7] infers the phylogenetic trees for a cohort of patients simultaneously and reconciles the heterogeneous trees using a matrix summarizing the frequencies of all pairwise ancestor-descendant relationships across tumors and outputs the trees having the smallest distance to the matrix. The entries in the normalized matrix are empirical estimates of the probability of one mutation being the ancestor of another mutation, which can be used to compute the probability of a possible evolutionary trajectory. HINTRA[8] extends the idea of REVOLVER by relaxing the assumption that the occurrence of a mutation depends only on its direct ancestor. It considers all possible sets of ancestors, allowing for more complex dependencies. However, the number of rows of the modified count matrix is exponential in the number of mutations, which limits the scalability of the algorithm[9]. Another direction is to find deterministic patterns from the mutation trees. RECAP solves an optimization problem that simultaneously clusters the patients into subgroups and selects one consensus tree for each cluster[9]. CONETT[10] and MASTRO[31] are computational methods to identify significantly conserved evolutionary trajectories by expanding the tumor trees into graphs that satisfy all partial order relations among the mutations. The former searches for a single conserved evolutionary trajectory tree that can describe the pattern in as many tumors as possible, whereas the latter outputs all such trajectory trees observed in at least a certain number of tumors. These deterministic methods do not provide probabilistic measures of future events given a trajectory or a tree and thus cannot be easily adapted for evolutionary predictions.

Here, we present a novel CPM, called TreeMHN, for simultaneous inference of patterns of clonal exclusivity and co-occurrence and repeated evolutionary trajectories from a cohort of intra-tumor phylogenetic trees. Unlike classical CPMs, including the genotype MHN method[30], TreeMHN considers the complete mutational histories of tumors and the dependencies between their subclonal structures represented by the intra-tumor mutation tree rather than the overall presence and absence of mutations. Compared to current state-of-the-art methods for detecting recurrent trajectories, TreeMHN is probabilistic and explicitly incorporates the exclusivity patterns of mutations. Using simulated data, we demonstrate the superior performance of TreeMHN in estimating the clonal exclusivity parameters and the probability distribution of evolutionary trajectories as compared to the genotype MHN method[30], REVOLVER[7], and HINTRA[8]. We then apply TreeMHN to three cancer cohorts: acute myeloid leukemia (AML)[21], non-small-cell lung cancer (NSCLC)[15], and breast cancer[32]. Our estimated exclusivity patterns and most probable evolutionary trajectories not only confirm previous biological findings but also provide new insights into the interdependencies of mutations, which could be informative for clinical decisions. With longitudinal tumor samples, TreeMHN provides improved predictions on the next mutational events given a tumor tree over alternative methods, which highlights the potential for evolution-based precision treatment plans.

## Results
### TreeMHN overview
We developed TreeMHN, a probabilistic model for inferring exclusivity patterns of mutations and recurrent evolutionary trajectories from tumor mutation trees. A tumor mutation tree $\mathscr{T}$ is a rooted tree that encodes the evolutionary history of a tumor[33]. The root corresponds to the start of the evolutionary process with no mutations (gray nodes in Fig. 1). The nodes represent the mutations connected according to the order in which they occur and fixate in the cell population (colored nodes in Fig. 1). Each path from the root to a node in the tree constitutes an evolutionary trajectory $\pi$, which characterizes the successive accumulation of mutations and uniquely defines a subclone. Starting from an existing subclone $\pi$, we assume that the time until a new mutation $i$ occurs and fixates, resulting in a new subclone $(\pi, i)$, follows an exponential distribution with its rate dependent on not only mutation $i$ but also all ancestor mutations in $\pi$. We parameterize the rates using a Mutual Hazard Network (MHN) $\Theta = \left(e^{\theta_{ij}}\right)_{i,j\in[n]} \in \mathbb{R}^{n \times n}$, where $n$ is the number of mutations. The diagonal entries $\{\Theta_{ii}\}_{i\in[n]}$ are the baseline rates of evolution, indicating how quickly each mutation will occur and fixate in a subclone independent of the other mutations. Mutations can have positive (co-occurring), negative (exclusive), or zero (no) effects on the rates of further downstream mutations. This is encoded by the off-diagonal entries $\{\theta_{ij}\}_{i\in[n]}$ with an equivalent graphical structure (Fig. 1). If $\theta_{ij}$ is positive, mutation $j$ increases the rate of mutation $i$ (denoted by an edge $j \rightarrow i$). If $\theta_{ij}$ is negative, mutation $j$ decreases the rate of mutation $i$, (denoted $j \dashv i$). The topology of a mutation tree is jointly determined by the waiting times of the subclones and an independent sampling time, which is also exponentially distributed. Only subclones acquired before the sampling time are observable in a tree. Therefore, the marginal probability of observing a tree $\mathscr{T}$ given $\Theta$ is equal to the probability that all observed mutational events happen before the sampling event, and all unobserved events that could happen next do not happen before the sampling event.

To perform inference with TreeMHN, the input is a set of $N$ independent tumor mutation trees with a total number of $n$ mutations, which can be constructed from bulk, multi-region, or single-cell sequencing data using phylogenetic methods. The output is $\hat{\Theta}$, an estimated MHN describing the exclusivity and co-occurrence patterns of mutations for the given cohort. With our efficient parameter estimation scheme based on regularized maximum likelihood estimation and a hybrid Monte Carlo expectation-maximization algorithm, the computational complexity depends on the maximum tree size, which is typically much smaller than the number of mutations. The number of parameters to estimate ($n^2$) often exceeds the number of observations ($N$). To prevent model overfitting, we can run TreeMHN with stability selection[34], where the parameters in $\Theta$ are estimated over many

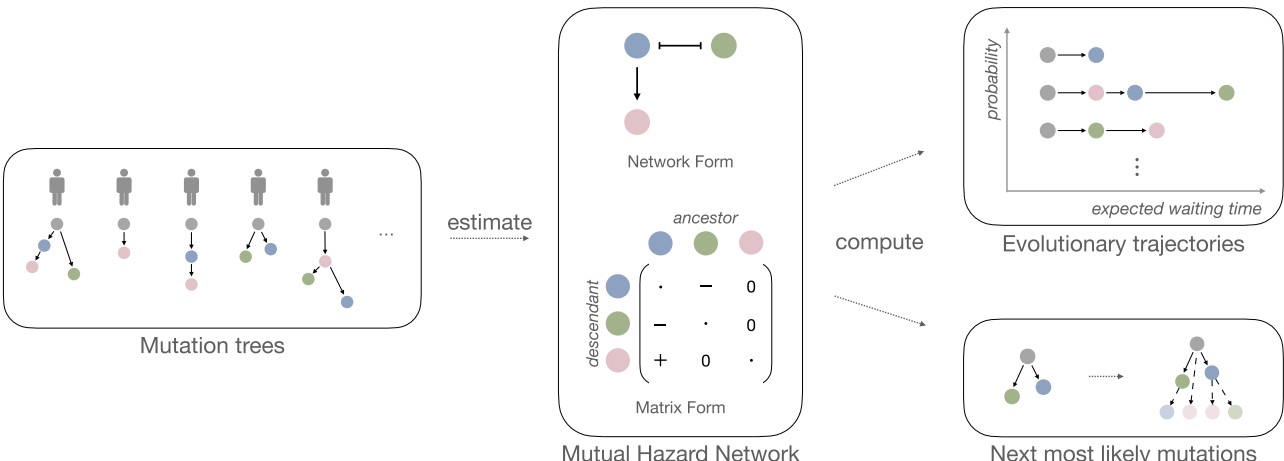

**Fig. 1 | Inference with TreeMHN.** The middle panel is a Mutual Hazard Network (MHN) for three distinct mutations (colored differently) represented as a network and equivalently as a matrix. The edges represent the co-occurring (→) or exclusive (⊣) stochastic dependencies among the mutations, corresponding to positive and negative off-diagonal entries in the matrix respectively. The diagonal entries are the baseline rates of fixation and have no influence on the network structure (therefore depicted as dots). Given a set of heterogeneous tumor mutation trees (left panel), we estimate the dependency parameters of an unknown underlying MHN. From the estimated MHN, we can compute the probability and the expected waiting time of any evolutionary trajectory (upper right panel). Additionally, we can compute the most probable next mutational events given an existing tree (lower right panel).

subsamples of the trees, and only those having a high probability of being non-zero are kept. This procedure can significantly improve the precision of identifying the true relationships among the mutations. Furthermore, the estimated model allows us to compute the probabilities of different evolutionary trajectories or evaluate the most likely next mutational events given a tumor tree. We provide an overview of the inference procedure in Fig. 1 and more technical details can be found in Methods and Supplementary Materials.

## Performance assessment on simulated data

Through simulations, we assess the performance of TreeMHN in comparison to alternative methods in estimating exclusivity patterns and associated distributions of evolutionary trajectories from mutation trees. For each simulation run, we randomly generate a ground truth network $\Theta$ with $n$ mutations and a set of $N$ mutation trees. Using the $N$ simulated trees as input, we run TreeMHN along with the genotype MHN method[30], REVOLVER[7], and HINTRA[8]. We consider different configurations of simulation experiments, including varying the number of mutations $n$ and the number of trees $N$. For each configuration, we perform 100 repetitions. More simulation details are provided in the Methods.

We first evaluate how well TreeMHN can estimate the patterns of clonal co-occurrence and exclusivity by computing the structural differences between the estimated network $\hat{\Theta}$ and the ground-truth network $\Theta$. Specifically, we measure the precision and recall of identifying the true edges in $\Theta$. An estimated off-diagonal entry in $\hat{\Theta}$ is a true positive if and only if it is non-zero and has the correct sign (Supplementary Fig. S3). We compare TreeMHN to the genotype MHN method, the only previously published cancer progression model that can estimate $\Theta$ (Fig. 2).

In all cases, TreeMHN clearly and consistently outperforms the genotype MHN approach in terms of both average precision and recall. This is because the problem of inferring an MHN from binary genotypes is in general underspecified, meaning that multiple different MHNs share similar likelihoods[35], which cannot be alleviated by taking subclonal genotypes directly as input. This result highlights the benefit of utilizing the existing ordering information encoded in the trees to resolve the underlying network. In fact, this benefit is maintained even if high levels of noise are added to the trees (Supplementary Figs. S5 and S6), as well as for varying proportions of zero entries (Supplementary Fig. S7) and negative entries in $\Theta$ (Supplementary Fig. S8). The

difference between the networks estimated using MLE and MC-EM is small and decreases with increasing number of Monte Carlo samples (Supplementary Fig. S9).

For TreeMHN and genotype MHN, we additionally implemented the stability selection procedure over a set of regularization parameters $\gamma \in \{0.05, 0.1, 0.5, 1, 1.5, 2, 2.5, 3\}$, where larger values of $\gamma$ promote sparser network structures and prevent model overfitting (Methods). The performance of both methods improves as the number of trees increases and the number of mutations decreases. At the same regularization level, the solutions with stability selection achieve much higher precision although at the cost of lower recall. By sorting the rows and columns of $\hat{\Theta}$ by descending baseline rates, we notice that both methods perform better in recovering the pairwise interactions between mutations with higher baseline rates than those with lower baseline rates (Supplementary Fig. S4). The reason is that interactions between rare mutations can hardly be observed in the trees, leading to a lack of statistical power to correctly estimate the corresponding entries in $\Theta$. This limitation is in general method-independent.

Next, we assess the performance of TreeMHN in estimating trajectory probabilities (Fig. 3). For comparison we also include REVOLVER and HINTRA, two probabilistic approaches for detecting repeated trajectories. Across different numbers of mutations and trees, TreeMHN outperforms all alternatives. HINTRA has the worst performance, possibly because of over-parameterization given the limited number of observations for every possible ancestry set. REVOLVER has similar performance as genotype MHNs even though it does not explicitly model co-occurrence and exclusivity between mutations. The genotype MHN method is unable to handle the case $n = 30$ due to exponentially increasing space and time complexity[30]. For TreeMHN, the runtime and memory are instead limited by the tree with the maximum number of subtrees (Supplementary Fig. S10).

## Application to acute myeloid leukemia data

We apply TreeMHN to the cohort of $N = 123$ AML patient samples analyzed in ref. 21 by high-throughput single-cell panel sequencing, which involves 543 somatic mutations in $n = 31$ cancer-associated genes and does not include any synonymous mutations. We assume that the mutation trees reconstructed by SCITE[33], a single-cell phylogenetic method, represent the complete evolutionary histories of the

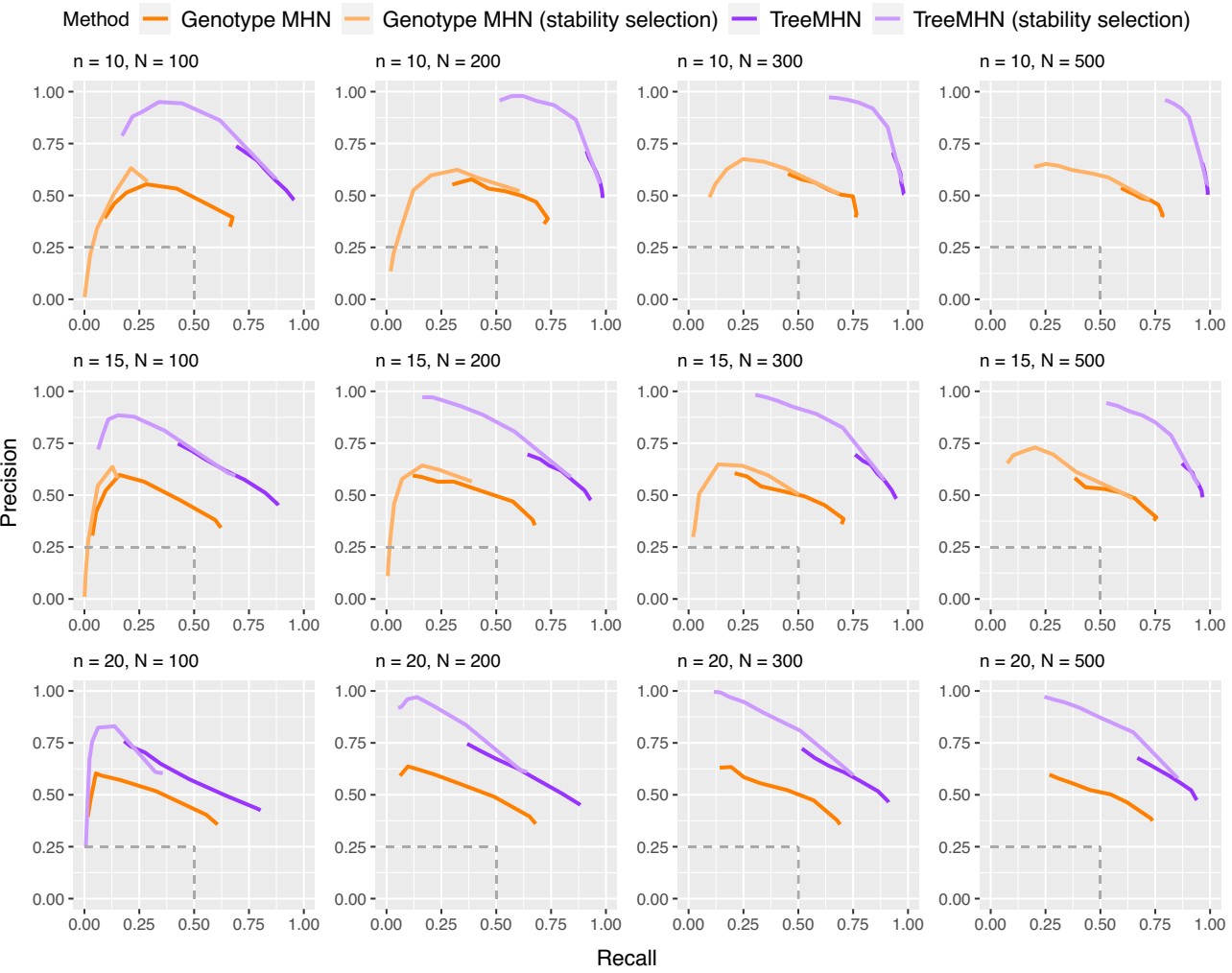

**Fig. 2 | Performance of TreeMHN and the genotype MHN method (with and without stability selection) in estimating the true network Θ on simulated data for *n* ∈ {10, 15, 20} mutations and *N* ∈ {100, 200, 300, 500} samples.** The precision and recall curves averaged over 100 simulation runs are plotted for the top half of mutations ranked by baseline rates. Each point on the curves corresponds to a penalization level *γ* ∈ {0.05, 0.1, 0.5, 1, 1.5, 2, 2.5, 3}. The dash lines indicate the performance of randomly guessing the edge directions in Θ. The curves for genotype MHN with stability selection are omitted for *n* = 20 due to excessive computation time. Source data are provided as a Source Data file.

tumors. Assuming all point mutations in the same gene have similar effects, we summarize the mutations at the gene level and observe many mutated genes appearing more than once in two separately evolved subclones of the same tree, such as *FLT3*, *KRAS*, and *PTPN11*[21] (Supplementary Section E.1). This recurrence increases the statistical power for detecting recurrent gene-level trajectories and patterns of clonal exclusivity.

To avoid overfitting, we train TreeMHN with stability selection and obtain a network over the 15 mutated genes which have non-zero off-diagonal entries (Fig. 4 and Supplementary Fig. S11). The sparseness of the network is because the number of observations for the pairwise interactions between rare events is so small that the corresponding entries are filtered out during stability selection. In this case, even if an interaction exists, there may not be enough power to detect it.

Among the 31 AML genes, *DNMT3A* has the highest baseline rate, followed by *IDH2*, *FLT3*, *NRAS*, *NPM1*, and *TET2*, which are known to be more frequently mutated in AML patients[36]. However, these genes do not necessarily appear in every mutation tree or are the initiating events, since the appearance of a mutation depends not only on the baseline rate but also on whether other mutations happened upstream with promoting or inhibiting effects, which

further explains the high degree of heterogeneity in the trees. By comparing the off-diagonal entries against the list of gene pairs found by GeneAccord[22] on the same dataset, we observe highly consistent but more informative results. In particular, most significant pairs of co-occurring or exclusive genes can be confirmed in the estimated network with additional directional strengths. For example, GeneAccord identifies *NRAS* and *FLT3* as significantly clonally exclusive. TreeMHN further reveals that the effect of *FLT3* inhibiting the occurrence of *NRAS* in the same lineage is much stronger than the other way around. In other words, if a subclone has already acquired mutations in *FLT3*, then it is less likely to accumulate another mutation in *NRAS*, which may still occur in separately-evolved subclones. On the contrary, subclones that acquire mutations in *NRAS* first are still relatively likely to hit *FLT3* subsequently. Apart from with *NRAS*, such a relationship also appears with *PTPN11* and *IDH2*. This observation aligns with previous studies[37], where some patients (e.g., 3/11 in ref. [38] and 15/41 in ref. [39]) developed secondary resistance to *FLT3* inhibitors due to off-target mutations (e.g., genes in the RAS pathways), which were present in small cell populations prior to treatment.

Next, we infer the most probable evolutionary trajectories from the estimated network (Methods and Fig. 5), which are consistent with

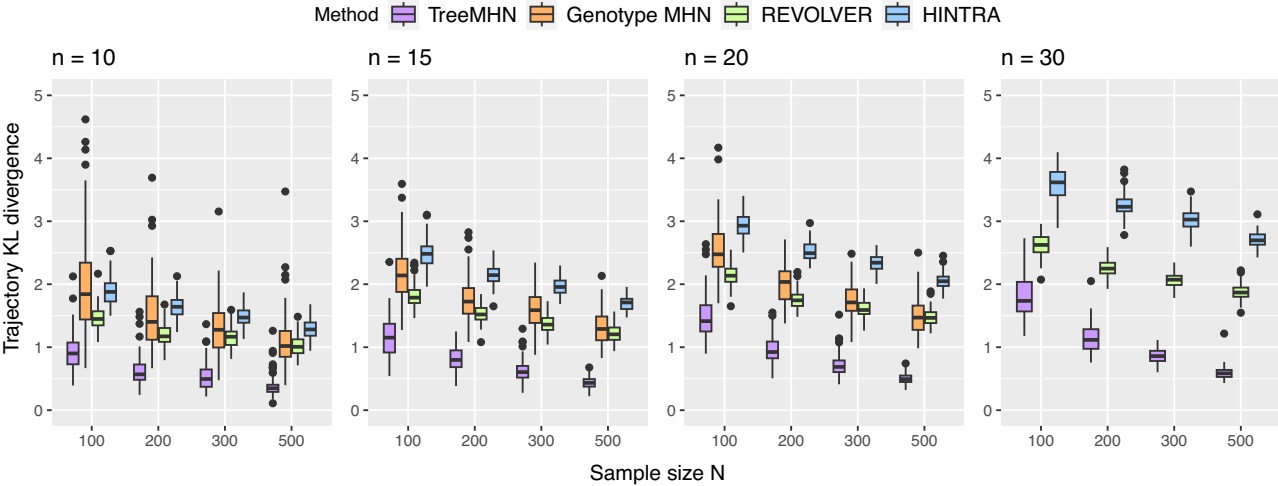

**Fig. 3 | Performance of TreeMHN in estimating the probability distribution of evolutionary trajectories compared to genotype MHNs, REVOLVER, and HIN-TRA (Methods).** We display the Kullback-Leibler divergence between the estimated distributions and the ground truth distributions for $n \in \{10, 15, 20, 30\}$ mutations and $N \in \{100, 200, 300, 500\}$ samples over 100 simulation runs. In the box plots, the box represents the interquartile range (IQR) with the median inside, while the whiskers extend to the minimum and maximum values within 1.5 times the IQR, and any data outside the whiskers are shown as individual points. Source data are provided as a Source Data file.

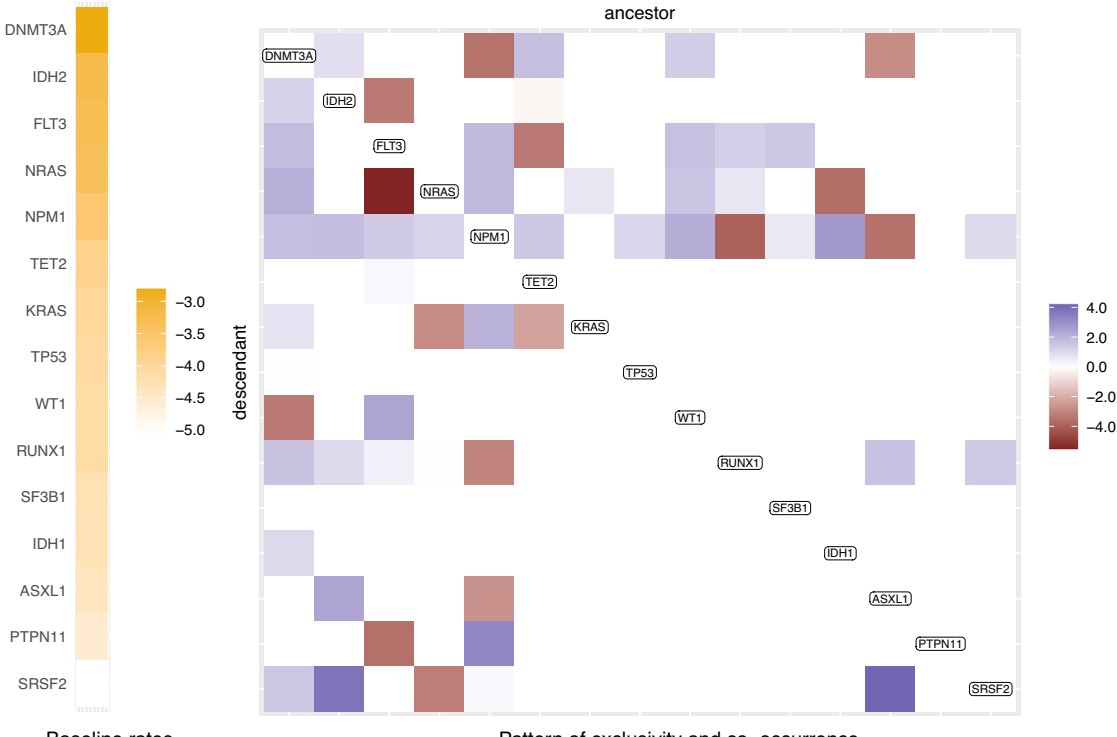

**Fig. 4 | Partial Mutual Hazard Network for the 123 AML patient samples.** The full network is shown in Supplementary Fig. S11, and only mutations with non-zero off-diagonal entries are shown here. The columns and rows of the matrix are ordered by decreasing baseline rates, which are shown on the left in yellow scale. The empty off-diagonal entries represent the case of no effect ($\theta_{ij} = 0$), meaning that there is no edge from mutation $j$ to mutation $i$. The red blocks correspond to inhibiting effects ($\theta_{ij} < 0, j \dashv i$) and the blue promoting effects ($\theta_{ij} > 0, j \rightarrow i$). The darker the colors, the stronger the effects. Source data are provided as a Source Data file.

the significantly conserved ones identified by CONETT[10] and MASTRO[31] (Supplementary Figs. S12 and S13). In comparison to alternative methods, the probabilities estimated by TreeMHN match closer in rankings with the relative frequencies of the observed AML trajectories (Supplementary Figs. S14 and S15). Mutations in *DNMT3A*, *NPM1*, and *FLT3* often co-occur with a relatively high probability. This three-way interaction is found to be associated with poor prognosis[40,41]. With

$DNMT3A \rightarrow NPM1$ and $NPM1 \dashv DNMT3A$, the ordering between them is more likely to be *DNMT3A* first followed by *NPM1*, which has been reported in previous studies[42,43].

Moreover, conditioned on the estimated network and a given tumor tree, we can predict the most probable next mutational event (Methods). To evaluate TreeMHN predictions, we perform both retrospective predictions on the rooted subtrees of the 123

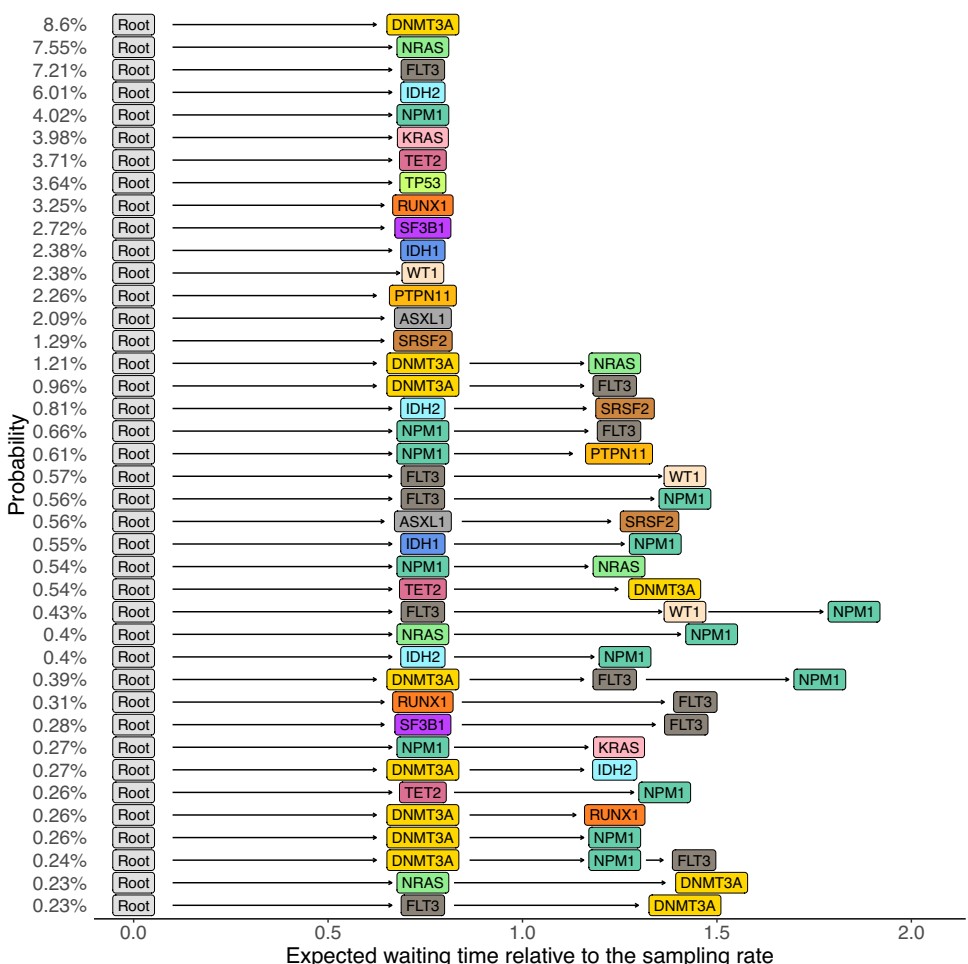

**Fig. 5 | Top 40 most probable evolutionary trajectories inferred from the partial Mutual Hazard Network of the AML dataset (Fig. 4).** Each row represents an evolutionary trajectory, labeled and ordered by trajectory probabilities, and the horizontal positions of the mutations correspond to their expected waiting times relative to the sampling rate of $\lambda_s = 1$ (Methods). Source data are provided as a Source Data file.

primary tumor trees and forward predictions using the longitudinal samples in the same dataset (Fig. 9). A comparative analysis against five alternatives (TreeMHN that estimates only the baseline rates of mutations, genotype MHN on consensus genotypes, genotype MHN on weighted subclonal genotypes, REVOLVER, and a frequency-based model that predicts the next events using the relative frequencies of the mutations in the cohort) reveals better predictive performance of TreeMHN. For the majority of the 123 primary tumor trees, TreeMHN assigns higher rankings to the events that actually happened downstream of any of the rooted subtrees (Fig. 6 & Supplementary Fig. S17). Under TreeMHN, seven out of nine new events in consecutive longitudinal samples have higher predicted probabilities than over 94% of the other possible mutational events, whereas alternative methods give varying predictions which are worse on average than TreeMHN (Table 1 and Supplementary Fig. S18). For instance, given the tree of patient sample AML-38-002, TreeMHN successfully identifies the two relapse events (Root → NPM1 → IDH2 → PTPN11 and Root → NPM1 → IDH1 → FLT3 in AML-38-003) in the top 5% most probable mutational events. With treatment effects drastically altering the fitness landscape[44], it is possible that subclones that were too small to detect at diagnosis became more abundant in a relapse. The consistent predictions highlight the ability of TreeMHN to unravel the interplay among the mutational events from the heterogeneous mutation trees, offering opportunities for evolution-guided treatment strategies.

## Application to non-small-cell lung cancer data

Next, we analyze the TRACERx NSCLC multi-region whole-exome sequencing data for $N = 99$ patients[15]. By applying TreeMHN to the phylogenetic trees with $n = 79$ driver mutations from ref. 7 (Supplementary Section F.1), we detect stable signals of interdependencies among 18 mutations (Fig. 7 and Supplementary Fig. S19). The majority of these interactions are from *KRAS*, *TP53*, and *EGFR* to other genes, suggesting their essential roles as recurrent initial events in tumorigenesis and progression[6,15]. In particular, the exclusive relationship between the oncogenic drivers *KRAS* and *EGFR* aligns with previous studies and could be associated with different clinical features (e.g., smoking exposure) between *KRAS*-mutant and *EGFR*-mutant subgroups[45,46]. Each of these two mutations is co-occurring with *TP53*, one of the most commonly mutated tumor suppressor genes in NSCLC[47]. *EGFR*-mutant tumors with co-occurring *TP53* were found to have higher degrees of genomic instability and shorter progression-free survival after EGFR TKI therapy[48]. Co-mutations in *KRAS/TP53*, on the other hand, are predictive for favorable clinical response to PD-1 inhibitors[49]. Moreover, existing *EGFR* mutations may encourage the occurrence of mutations in *TERT* and *PIK3CA*, where the former can influence telomere maintenance mechanisms in tumor cells[50], and the latter promotes cellular invasion and migration in vitro[46]. Both are associated with poor overall survival[46,50]. These observations indicate that the tumor progression processes leading to the observed co-occurrence of the genes may have direct clinical consequences.

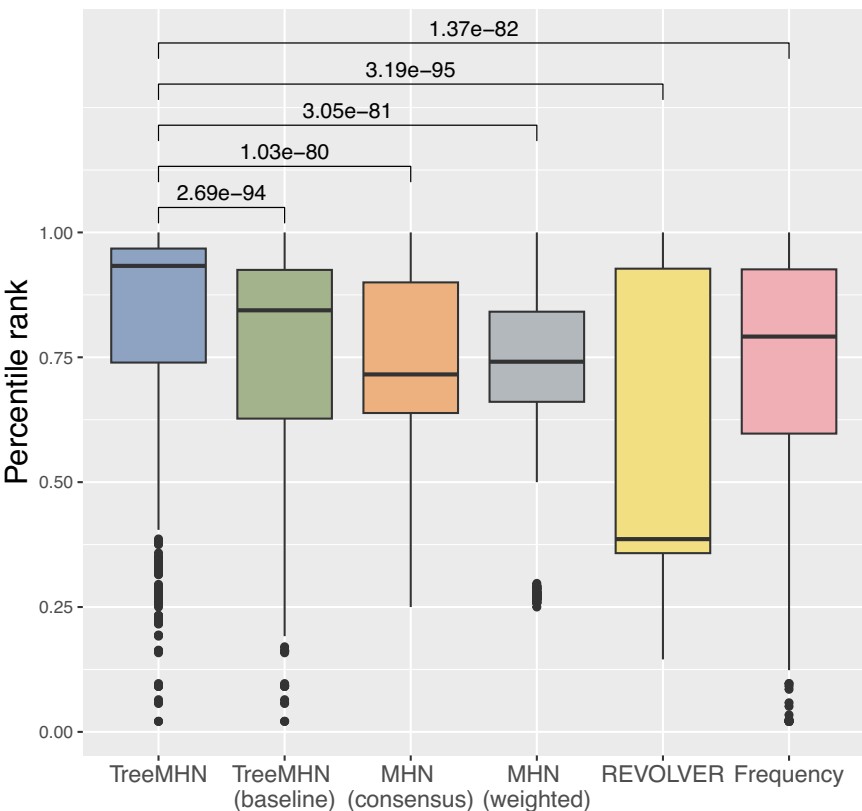

**Fig. 6 | Performance assessment on retrospective predictions for the AML dataset.** Each boxplot shows the percentile ranks computed using TreeMHN and alternative methods on the 1370 pairs of rooted subtrees and their corresponding downstream events, enumerated from the 123 independent primary tumor trees. The box represents the interquartile range (IQR) with the median inside, while the whiskers extend to the minimum and maximum values within 1.5 times the IQR, and any data outside the whiskers are shown as individual points. The p-values with Bonferroni correction for the two-sided Wilcoxon signed-rank tests between TreeMHN and alternative methods are also displayed. Source data are provided as a Source Data file.

Without reconciling the trees or clustering the trees into subgroups, the most probable evolutionary trajectories inferred by TreeMHN capture most of the repeated trajectories found by REVOLVER (Supplementary Fig. S20). Notably, we recover 7 of the 10 REVOLVER clusters (C2, C3, C4, C6, C7, C9, C10) in the top 50 trajectories (Supplementary Fig. S21), underlining the ability of TreeMHN to disentangle noisy signals. For cluster C7, TreeMHN assigns a higher probability to the trajectory $KRAS \rightarrow TP53 \rightarrow MGA$ as compared to $TP53 \rightarrow KRAS \rightarrow MGA$, whereas REVOLVER cannot tell them apart. Interestingly, the trajectory $CDKN2A \rightarrow TP53$ in cluster C5, identified as highly robust by REVOLVER, has a lower probability than $TP53 \rightarrow CDKN2A$, where the ordering is reversed. As pointed out in ref. 15, most of the mutations in TP53 were clonal for both adenocarcinoma and squamous-cell carcinoma in the TRACERx dataset, whereas mutations in CDKN2A appeared mostly in the latter subtype and often late. Hence, it is more likely that mutations in TP53 precede those in CDKN2A, hinting at how they cooperate on cell-cycle deregulation.

**Application to breast cancer data**

Finally, TreeMHN is not restricted to tumor trees reconstructed from single-cell or multi-region sequencing data, and we analyze a bulk sequencing dataset of 1918 tumors for 1756 advanced breast cancer patients, where clinical data is also available[32]. Based on hormone receptor and HER2 status (HR+/HER2+, HR+/HER2-, HR-/HER2+, Triple Negative) and sample type (treatment-free primary vs. metastasis), we can segregate the patients into eight subgroups. Considering only copy-neutral autosomal regions, we restrict the analysis to the union of SNVs that appear in at least 10% of the patients in each subgroup. In total we have $n = 19$ mutations and $N = 1152$ patients with 1232 phylogenetic trees inferred by SPRUCE[51], a phylogenetic method based on bulk mixture deconvolution (Supplementary Section G.1).

The estimated network on all trees captures combined signals, which are mainly driven by the largest subgroups with HR+/HER2- status (Supplementary Fig. S22). We observe that mutations in TP53 and PIK3CA, which were identified as early clonal events[6,8,32], have the highest baseline rates of mutations and are co-occurring. We also detect exclusivity between PIK3CA and PIK3R1, suggesting that mutations in one of these drivers may suffice to trigger abnormal regulation of PI3K activation in breast cancer[52]. While there are no large deviations from the combined network, we obtain some subgroup-specific interdependencies by applying TreeMHN separately to each subgroup (Supplementary Figs. S23 and S24). For example, NF1 mutations, which are found to be associated with endocrine resistance[32], have higher baseline rates in hormone receptor-negative tumors and co-occur with TP53 more frequently in metastatic samples. Also, we detect co-occurrence of PIK3CA and GATA3 in hormone receptor-positive tumors but not in the negative ones. It has been observed that tumors with GATA3 mutations depend more on estrogen level and may be predictive for positive response to aromatase inhibitors[53]. The subgroup-level most probable evolutionary trajectories not only confirm existing findings reported by HINTRA[8] and RECAP[9], such as $CDH1 \rightarrow PIK3CA$ for the HR+/HER2- samples, but also provide new insights (Supplementary Figs. S25–S28). For instance, the combination of TP53, PTEN, and PIK3CA/PIK3R1 ranks higher in the metastatic triple-

**Table 1 | Table of estimated percentile ranks for the new events in consecutive tree pairs in the longitudinal samples of the AML dataset**

| From | To | New Event | Estimated Percentile Ranks | | | | | | | Frequency |
|---|---|---|---|---|---|---|---|---|---|---|
| | | | TreeMHN | TreeMHN (baseline) | MHN (consensus) | MHN (weighted) | REVOLVER | | |
| AML-63-004 | AML-63-003 | Root → IDH2 → NPM1 | 98.33 | **99.17** | 63.33 | 96.67 | 98.33 | 95.83 |
| AML-38-002 | AML-38-003 | Root → NPM1 → IDH2 → PTPN11 | **97.61** | 71.71 | 69.72 | 72.91 | 83.86 | 60.96 |
| AML-99-003 | AML-99-004 | Root → DNMT3A → IDH2 → RUNX1 → FLT3 | **97.32** | 94.94 | 65.77 | 90.77 | 95.54 | 95.54 |
| AML-83-001 | AML-83-002 | Root → DNMT3A → IDH2 → NRAS | **97.16** | 92.05 | 87.5 | 93.18 | 95.45 | 92.05 |
| AML-09-001 | AML-09-002 | Root → NPM1 → FLT3 → WT1 | 95.69 | 78.02 | 80.6 | 81.47 | **100** | 75.43 |
| AML-38-002 | AML-38-003 | Root → NPM1 → IDH1 → FLT3 | 95.42 | **98.8** | 96.41 | 91.63 | 39.44 | 93.63 |
| AML-99-003 | AML-99-004 | Root → DNMT3A → IDH2 → NRAS | **94.64** | 91.37 | 91.37 | 94.35 | **94.64** | 90.48 |
| AML-04-001 | AML-04-002 | Root → SF3B1 → SRSF2 → NRAS → WT1 | 73.64 | 76.14 | **81.14** | 76.14 | 35.91 | 72.73 |
| AML-04-001 | AML-04-002 | Root → SF3B1 → SRSF2 → IDH1 | 70.23 | 67.05 | 76.36 | 69.32 | **83.41** | 65 |
| | | Mean | **91.12** | 85.47 | 79.14 | 85.16 | 80.73 | 82.4 |
| | | Median | **95.69** | 91.37 | 80.6 | 90.77 | 94.64 | 90.48 |
| | | (1st quantile, 3rd quantile) | **(94.64, 97.32)** | (76.14, 94.94) | (69.72, 87.50) | (76.14, 93.18) | (83.41, 95.54) | (72.73, 93.63) |

The highest percentile rank across TreeMHN and alternative methods for each row is in bold. Source data are provided as a Source Data file.

negative subgroup than in other subgroups. Cooperatively, these mutations can lead to hyperactivation of the PI3K-AKT pathway and uncontrolled proliferation of cells, ultimately reducing the overall survival rate[54]. Therefore, TreeMHN is capable of extracting key patterns related to clinical outcome from highly heterogeneous tumor mutation histories.

## Discussion

We have developed TreeMHN, a novel cancer progression model for the joint inference of repeated evolutionary trajectories and patterns of clonal co-occurrence or exclusivity from a cohort of intra-tumor phylogenetic trees. Unlike Mutual Hazard Networks[30], TreeMHN can take as input heterogeneous tree structures estimated from multi-region, single-cell, or bulk sequencing data, rather than per-tumor or per-clone consensus genotypes. Importantly, with our efficient parameter estimation procedure, it is the maximum tree size, rather than the total, typically much larger, number of mutations, that limits the computation time of TreeMHN.

Through simulation studies, we have demonstrated the superior performance of TreeMHN in estimating both patterns of clonal exclusivity or co-occurrence and trajectory probabilities in comparison with MHNs, REVOLVER, and HINTRA. Moreover, we have shown that TreeMHN is robust against uncertainty in the phylogenetic trees for varying noise levels. Alternatively, one may handle such uncertainty by sampling the trees for each patient proportionally to their posterior probabilities and taking them as weighted inputs to TreeMHN. By exploiting the evidence of temporal ordering among mutations contained in the tree topologies and properly accounting for clonal dependencies, TreeMHN can better resolve the underlying network structure. Given the estimated parameters, TreeMHN allows us to compute the probabilities of different evolutionary trajectories and the expected waiting times between mutational events. However, in general, these waiting times cannot be interpreted as real calendar time since they are with respect to the unknown sampling times and the scaling factor is therefore unknown. One remedy is to use longitudinal data, where the sampling time is either provided or can be inferred from data. Including observed sampling times is technically possible[55], but such data are often difficult to obtain without having treatment interventions. Modeling drug response data is a crucial but challenging direction to explore, for which TreeMHN may serve as a basis.

Unlike REVOLVER and HINTRA, our method embraces the heterogeneity among the trees and incorporates clonal exclusivity and co-occurrence into the analysis of recurrent evolutionary processes. Also, TreeMHN does not rely on any particular phylogenetic method. It is possible to combine different phylogenies from various sources (e.g., refs. [33,56,57]) to take into account different modeling assumptions. Future developments in phylogenetic methods together with more available data can further improve the estimate of TreeMHN. Another advantage of TreeMHN is the ability to model parallel mutations in distinct lineages, which are not uncommon in real data[21], while many of the existing alternatives require the infinite sites assumption. Like all other progression models, however, TreeMHN currently does not consider back mutations, i.e., situations in which a mutation is acquired at first but subsequently lost[58,59]. A possible extension along this line is to include additional parameters and use as input phylogenetic trees inferred by methods such as SCARLET[60], which views a decrease in copy numbers that overlap a mutated locus as evidence of back mutations. Moreover, TreeMHN is not designed for a specific type of mutation, such as SNVs. In other words, it is possible to detect recurrent trajectories at the level of copy number alterations[61], mixed types of mutations[62], or even functional pathways[63].

Nevertheless, TreeMHN does not take into account the subclone sizes, which can be viewed as consequences of clonal selection[64]. As noted by ref. [65], inferring mutation rates and fitness values jointly may

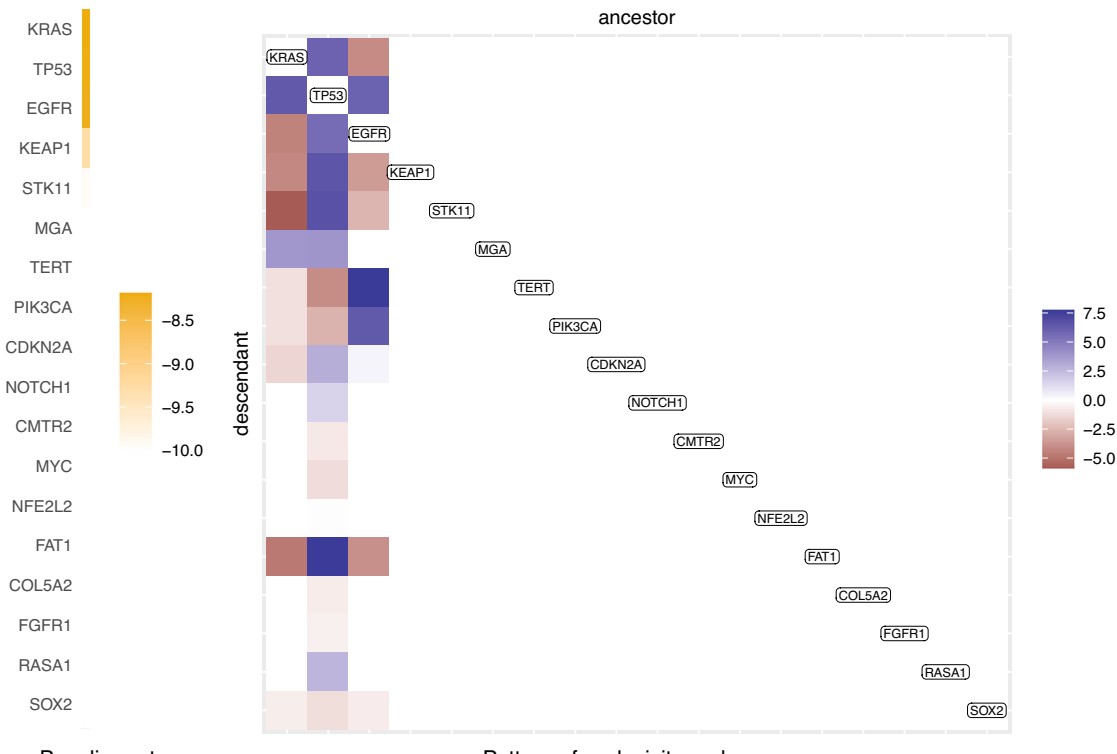

**Fig. 7 | Partial Mutual Hazard Network for the 99 TRACERx NSCLC patient samples.** The full network is shown in Supplementary Fig. S19, and here only mutations with non-zero off-diagonal entries are shown. The columns and rows of the matrix are ordered by decreasing baseline rates. Source data are provided as a Source Data file.

have promise but is challenging. On the one hand, larger subclone sizes can be attributed to both their earlier appearance or higher fitness[65]. On the other hand, the mutation rates in cancer progression models are rates of evolution, which implicitly involve subclone fitness. Thus, any attempts to modify TreeMHN to model clonal selection need to be taken with caution. In this work, we assumed all trees in the same cohort are generated from one MHN, but it is possible to extend TreeMHN to perform model-based unsupervised clustering akin to ref. [66].

We have applied TreeMHN to data of three different cancer types and demonstrated its practicality and flexibility. Beyond reproducing existing results, TreeMHN provides additional insights, including baseline rates of mutational events, directional strengths of the promoting or inhibiting effects, as well as probabilistic measures of evolutionary trajectories. In particular, we find good overlap between the longitudinal AML data[21] and our predictions of the next mutational events. The application and results of TreeMHN may therefore be useful for prioritizing personalized treatments.

## Methods

### TreeMHN generative model

TreeMHN models mutation trees using a tree-generating process. Consider $N$ mutation trees $\mathscr{T} = \{\mathscr{T}^1, \ldots, \mathscr{T}^N\}$ with a total number of $n$ mutations. Each tree $\mathscr{T}^l$ corresponds to the mutational history of a tumor and contains a subset of mutations from $[n] := \{1, \ldots, n\}$. We assume that the trees are realizations of a Markov process with the transition rate matrix parameterized by a Mutual Hazard Network $\Theta \in \mathbb{R}^{n \times n}$. We denote each subclone in a tree by the evolutionary trajectory $\pi$ that runs from the root 0 to the node where the subclone is attached. In other words, a subclone $\pi$ is a sequence $(0, \sigma_1, \ldots, \sigma_d)$ with $0 \leq d \leq n$, and $\sigma_i \in [n]$ are non-duplicated elements. Let $\Pi$ denote the space of all subclones, or equivalently, the space of all evolutionary

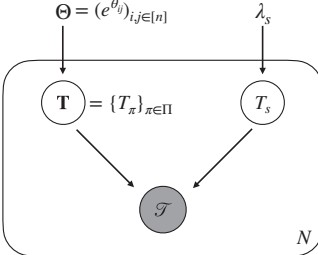

**Fig. 8 | TreeMHN as a probabilistic graphical model.** The waiting times of subclones $\mathbf{T} = \{T_\pi\}_{\pi \in \Pi}$ are exponentially distributed random variables parameterized by an MHN $\Theta = (e^{\theta_{ij}})_{i,j \in [n]}$. The tree structure $\mathscr{T}$ is jointly determined by $\mathbf{T}$ and an independent sampling time $T_s$, which is also an exponential random variable with rate $\lambda_s$. Both $\mathbf{T}$ and $T_s$ are hidden variables. The random variables inside the plate replicate $N$ times, which generate $N$ independent and identically-distributed mutation trees.

trajectories. The tree-generating process is defined as follows (Fig. 8, Supplementary Sections A.1 and A.2):

1. The initial wild-type subclone is $\pi = (0)$ at time $T_{(0)} = 0$. For each subclone $\pi$, the set of mutations that could happen next is $[n] \setminus \pi$.
2. The waiting time until a new subclone $(\pi, i)$ with $i \in [n] \setminus \pi$ is born from $\pi$ is an exponentially distributed random variable,

$$T_{(\pi,i)} \sim T_\pi + \mathrm{Exp}(\lambda_{(\pi,i)}), \qquad \lambda_{(\pi,i)} = \exp\left(\theta_{ii} + \sum_{j \in \pi} \theta_{ij}\right) = \Theta_{ii} \prod_{j \in \pi} \Theta_{ij} \quad (1)$$

where $\theta_{ii} = \log \Theta_{ii}$ is the baseline rate of mutation $i$, and $\theta_{ij} = \log \Theta_{ij}$ determines the positive (denoted $j \rightarrow i$), negative ($j \dashv i$) or zero effect of

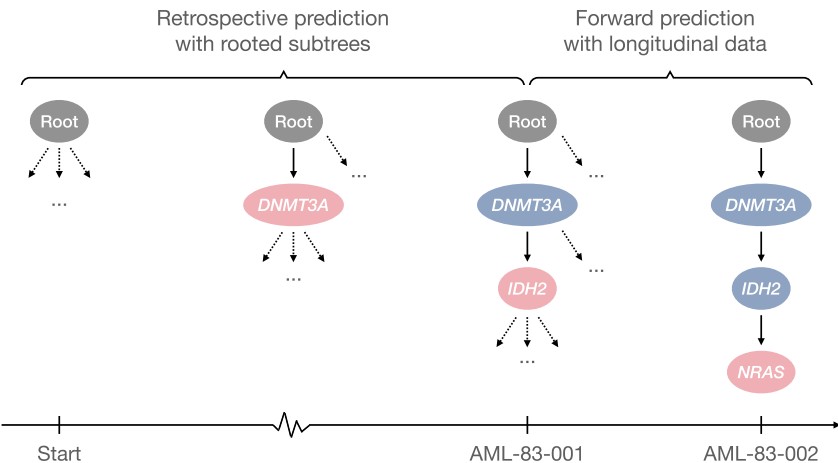

**Fig. 9 | A schematic representation of evaluating the predictions of next events given a tree structure.** AML-83-001 is a tumor sample, and the associated mutation tree is a chain with two mutations. Given this tree, we can retrospectively enumerate all its rooted subtrees and compute the probability of the existing downstream events. AML-83-002 is a consecutive sample available for the same patient, with which we can perform forward prediction and validate the result. The root node is in gray, the events that are common from a preceding tree are in blue, and the new events are in pink.

an existing mutation $j$ on mutation $i$[30]. If both $\theta_{ij} < 0$ and $\theta_{ji} < 0$ (i.e., $j\dashv\vdash i$), then the two mutations are called clonally exclusive, and if both signs are positive (i.e., $j \leftrightarrow i$), we say that the mutations are co-occurring. The collection of all waiting times is denoted by $\mathbf{T} = \{T_\pi\}_{\pi \in \Pi}$.

3. The observed tree structure $\mathcal{T}$ is co-determined by an independent sampling event $S$ with time $T_s \sim \mathrm{Exp}(\lambda_s)$. Typically, we assume $\lambda_s = 1$ for identifiability[29]. An edge from $\pi$ to $(\pi, i)$ exists in $\mathcal{T}$ if and only if $T_{(\pi, i)} < T_s$, i.e., if mutation $i$ happened before sampling the tumor cells.

4. The process iterates until all subclones that could emerge next have a longer waiting time than the sampling time. We denote the augmented tree structure by $A(\mathcal{T})$, which includes the edges pointing towards the events right after sampling (Supplementary Fig. S1b). Events further downstream are not considered as they cannot influence the observed tree structure.

One advantage of this formulation is that the same mutational event can appear in different lineages of a tree. For computational convenience, many of the alternatives capable of inferring repeated evolutionary trajectories (e.g., REVOLVER, HINTRA, RECAP) require the infinite sites assumption (ISA), i.e., each mutational event (e.g., an SNV) is gained at most once. However, it has been shown that allowing parallel mutations is, in general, a more realistic assumption[58]. Even assuming the ISA for individual genomic bases, when summarizing at the gene level the same gene may be affected in parallel lineages. In results, we demonstrated the applicability of TreeMHN using the AML dataset which contains parallel mutations. Due to the introduction of clonally exclusive mutations, another advantage is the possibility to generate very distinct trees from given mutual hazards. Therefore, TreeMHN can model extensive intra-tumor and inter-tumor heterogeneity, while capturing common, re-occurring features.

**Parameter estimation**
The marginal probability of observing a tree $\mathcal{T}$ conditioned on an MHN $\Theta$ is given by (Supplementary Fig. S1),

$$p(\mathcal{T}|\Theta) = P\left( \max_{\pi \in \mathcal{T}} T_\pi < T_s < \min_{\substack{\pi' \notin \mathcal{T} \\ \pi' \in A(\mathcal{T})}} T_{\pi'} \mid \Theta \right). \tag{2}$$

Both $p(\mathcal{T}|\Theta)$ and its gradients $\partial p(\mathcal{T}|\Theta)/\partial \lambda_\pi$ can be computed efficiently by inverting specific triangular matrices, which are constructed using the rates associated with the events in $A(\mathcal{T})$, and the

dimensions depend only on the number of subtrees in $\mathcal{T}$ (see Theorem 1 and 2 in Supplementary Section A.2 and the derivations in Supplementary Section A.3). Given $N$ mutation trees $\mathcal{T} = \{\mathcal{T}^1, \ldots, \mathcal{T}^N\}$, we follow[30] and estimate $\Theta$ by maximizing the penalized log-likelihood,

$$\hat{\Theta} = \arg\max_\Theta \left( \sum_{l=1}^N \log p(\mathcal{T}^l|\Theta) - \gamma \sum_{i \neq j} |\log \Theta_{ij}| \right), \tag{3}$$

where $\gamma > 0$ controls the sparsity of $\Theta$ in order to mitigate overfitting.

When some or all of the trees in $\mathcal{T}$ have many subtrees (e.g., more than 500 subtrees), the MLE procedure can still be very slow or even infeasible. In this case, we resort to a hybrid EM and Monte Carlo EM algorithm based on importance sampling as follows (see also Supplementary Section A.4). In the E step, given the observed trees $\mathcal{T}$ and the current estimate $\Theta^{(k)}$ for iteration $k \in \{0, 1, 2, \ldots\}$, we compute the expected value of the complete-data log-likelihood with respect to the unobserved collection of waiting times as

$$g(\Theta, \Theta^{(k)}) = \sum_{l=1}^N \sum_{i=1}^n \sum_{\pi \in \mathcal{T}^l : (\pi, i) \in A(\mathcal{T}^l)}$$
$$\left[ \log \Theta_{ii} + \sum_{j \in \pi} \log \Theta_{ij} - \Theta_{ii} \prod_{j \in \pi} \Theta_{ij} \mathbb{E}_{\mathbf{T}^l, T_s^l | \mathcal{T}^l, \Theta^{(k)}} (T_{(\pi, i)}^l - T_\pi^l) \right] + C, \tag{4}$$

where $C$ is a constant. For small trees, we can calculate the expected time difference in exact form,

$$\mathbb{E}_{\mathbf{T}, T_s | \mathcal{T}, \Theta}[T_{(\pi, i)} - T_\pi] = \frac{1}{\lambda_{(\pi, i)}} - \frac{1}{P(\mathcal{T}|\Theta)} \frac{\partial P(\mathcal{T}|\Theta)}{\partial \lambda_{(\pi, i)}}. \tag{5}$$

For large trees, we approximate the expectation by drawing $M$ samples from the following proposal distribution. First, the sampling time $T_s \sim \mathrm{Exp}(\lambda_s)$ with $\lambda_s = 1$ is drawn independently. Using the equation $T_{(\pi, i)} = T_\pi + Z_{(\pi, i)}$, we then follow the topological order in $\mathcal{T}$ and sample the difference in waiting times between subclones $\pi$ and $(\pi, i)$ recursively as

$$Z_{(\pi, i)} \sim \begin{cases} \mathrm{TExp}(\lambda_{(\pi, i)}, 0, t_s - t_\pi) & \text{if } (\pi, i) \in \mathcal{T} \\ \mathrm{TExp}(\lambda_{(\pi, i)}, t_s - t_\pi, \infty) & \text{if } (\pi, i) \in A(\mathcal{T}) \setminus \mathcal{T} \end{cases} \tag{6}$$

where $\mathrm{TExp}(\lambda, a, b)$ is the truncated exponential distribution with parameter $\lambda$ and bounds $0 \leq a < b < \infty$. Thus, the time points $\boldsymbol{t}^{(m)}$ and $t_s^{(m)}$

for $m = 1, \ldots, M$ generated from our proposal distribution are by definition compatible with the tree structure. The approximation is then

$$
\mathbb{E}_{T, T_s | \mathscr{T}, \Theta}[T_{(\pi, i)} - T_\pi] \approx \frac{\frac{1}{M} \sum_{m=1}^{M} w^{(m)}(t_{(\pi,i)}^{(m)} - t_\pi^{(m)})}{\frac{1}{M} \sum_{m=1}^{M} w^{(m)}},
$$

$$
w^{(m)} = \prod_{i=1}^{n} \left[ \prod_{\substack{(\pi, i) \in \mathscr{T}}} (1 - e^{-\lambda_{(\pi,i)}(t_s^{(m)} - t_\pi^{(m)})}) \times \prod_{\substack{(\pi, i) \notin \mathscr{T} \\ (\pi, i) \in A(\mathscr{T})}} e^{-\lambda_{(\pi,i)}(t_s^{(m)} - t_\pi^{(m)})} \right]. \quad (7)
$$

In the M step, we update $\Theta$ by maximizing the penalized expected complete-data log-likelihood

$$
g(\Theta, \Theta^{(k)}) - \gamma \sum_{i \neq j} |\log \Theta_{ij}|. \quad (8)
$$

To prevent overfitting, we run TreeMHN with stability selection[34] such that the parameters in $\Theta$ are estimated over many subsamples of the trees $\mathscr{T}$, and only those having a high probability of being non-zero are kept (see also Supplementary Section B). The choice of parameter estimation method is automatically determined by a pre-specified number of subtrees: if the maximum number of subtrees of all trees in the set is below this threshold, then MLE is used. Otherwise, we use the hybrid MC-EM method instead. By default, the threshold is set to be 500 subtrees, and the number of Monte Carlo samples is $M = 300$.

## Probability and expected waiting time of a trajectory

We consider the set of evolutionary trajectories that end with the sampling event $S$,

$$
\Pi_S : = \{(0, \sigma_1, \ldots, \sigma_d, S) | \sigma_i \in [n] \text{ non-duplicated}, 0 \leq d \leq n\}. \quad (9)
$$

Given $\Theta$ and $\lambda_s$, we can compute the probability of a trajectory $\pi \in \Pi_S$ as the product of competing exponentials,

$$
P_\Theta(\pi) = \left( \prod_{i=1}^{d} \frac{\lambda_{(\pi_{i-1}, \sigma_i)}}{\lambda_s + \sum_{j \in [n] \setminus \pi_{i-1}} \lambda_{(\pi_{i-1} j)}} \right) \times \frac{\lambda_s}{\lambda_s + \sum_{j \in [n] \setminus \pi} \lambda_{(\pi_j)}}, \quad (10)
$$

where $\pi_i = (0, \sigma_1, \ldots, \sigma_i) \subset \pi$. It follows that $(P_\Theta(\pi))_{\pi \in \Pi_S}$ is the probability distribution over $\Pi_S$ with respect to $\Theta$, since $\sum_{\pi \in \Pi_S} P_\Theta(\pi) = 1$. Likewise, the expected waiting time of $\pi \in \Pi_S$ is the sum of the expected time interval lengths along the trajectory,

$$
\mathbb{E}_\Theta[T_\pi] = \left( \sum_{i=1}^{d} \frac{1}{\lambda_s + \sum_{j \in [n] \setminus \pi_{i-1}} \lambda_{(\pi_{i-1} j)}} \right) + \frac{1}{\lambda_s + \sum_{j \in [n] \setminus \pi} \lambda_{(\pi_j)}}. \quad (11)
$$

However, computing $P_\Theta(\pi)$ for all $\pi \in \Pi_S$ is computationally infeasible for large $n$, since the number of trajectories in $\Pi_S$ is factorial in $n$. Instead, we can enumerate the most probable trajectories that have at least one mutation with dynamic programming following the pseudocode in Supplementary Algorithm 2, the complexity of which is linear in $n$.

To compare with alternative methods that do not contain a sampling event, such as REVOLVER[7] and HINTRA[8], we can use another formulation with $\Pi_d \subseteq \Pi$, which is the set of evolutionary trajectories of a fixed length $d$ for $1 \leq d \leq n$. The probability of a trajectory $\pi \in \Pi_d$ can be computed similarly as

$$
P_\Theta(\pi) = \prod_{i=1}^{d} \frac{\lambda_{(\pi_{i-1}, \sigma_i)}}{\sum_{j \in [n] \setminus \pi_{i-1}} \lambda_{(\pi_{i-1} j)}}, \quad (12)
$$

and $\sum_{\pi \in \Pi_d} P_\Theta(\pi) = 1$. Note that in this formulation, trajectories of different lengths are not directly comparable, but it is still useful as a tool to validate an estimated trajectory probability distribution. Suppose $Q$ is another probability distribution over $\Pi_d$, then the Kullback-Leibler

(KL) divergence from $Q$ to $P_\Theta$,

$$
D_{KL}(P_\Theta \| Q) = \sum_{\pi \in \Pi_d} P_\Theta(\pi) \log \frac{P_\Theta(\pi)}{Q(\pi)} \quad (13)
$$

measures the distance between the two distributions. In particular $D_{KL}(P_\Theta \| Q) = 0$ if and only if $P_\Theta = Q$.

## Probability of a downstream event given a tree

Given a tree structure $\mathscr{T}$ and an estimated network $\Theta$, we can compute the probabilities of the next mutational events. The events that could happen next are all events in the augmented tree $A(\mathscr{T})$ but not in $\mathscr{T}$. By competing exponentials, we calculate the probability of an event $\pi$ happens before all the other events in $A(\mathscr{T}) \setminus \mathscr{T}$ as

$$
p(\pi | \mathscr{T}, \Theta) = \frac{\lambda_\pi}{\sum_{\pi' \in A(\mathscr{T}) \setminus \mathscr{T}} \lambda_{\pi'}} \times \mathbb{1}\{\pi \in A(\mathscr{T}) \setminus \mathscr{T}\}, \quad (14)
$$

where $\lambda_\pi$ is the rate associated with event $\pi$ (Eq. (1)). Then, we can rank all events $\pi \in A(\mathscr{T}) \setminus \mathscr{T}$ by their probabilities. We evaluate TreeMHN predictions on the AML dataset[21]. We perform both retrospective predictions on the rooted subtrees of the 123 primary tumor trees and forward predictions using the longitudinal samples from 15 patients in the same dataset (Fig. 9). For retrospective predictions, we first take a primary tumor tree and exclude it from training. Then, we enumerate all its rooted subtrees and their corresponding downstream events, followed by computing the probability of each downstream event conditioned on a parent subtree and the estimated network (Eq. (14)). For forward predictions, on the other hand, we take the longitudinal samples and consider only those tree pairs on consecutive time points, where the second tree has mutations not present in the first tree. We compute the probabilities for the new events using the matrix estimated from the 123 primary tumor trees (i.e., longitudinal samples are excluded from training). While TreeMHN is unique in this task, we adapt five alternative methods for comparison (see also Supplementary Section C.2):

- TreeMHN (baseline): Here, we run TreeMHN with the restriction that all off-diagonal elements in the estimated network are zero and keep only the baseline rates. This approach assumes that all mutations are independent of each other while respecting the tree structures. The way to compute the probabilities of the next mutational events given a tree stays the same.
- MHN (consensus): We estimate $\hat{\Theta}_{MHN}$ using the genotype MHN method[30] on the consensus genotypes of the patient samples. A consensus genotype of a tumor is obtained by keeping the mutations that appear in more than 50% of the cell population. Given $\hat{\Theta}_{MHN}$, we compute the probabilities of the next events using Eq. (14).
- MHN (weighted): This method is the same as MHN (consensus) except for using the subclonal genotypes weighted by the subclone sizes as input to train $\hat{\Theta}_{MHN}$.
- REVOLVER: We first convert the subclonal genotypes into cancer cell fractions, the proportion of cancer cells having a specific mutation, followed by running REVOLVER to obtain the information transfer matrix $w$ (Supplementary Fig. S16). An entry $w_{ij}$ is the number of times mutation $i$ occurs before mutation $j$ in the patients. By row-normalizing $w$, we can obtain the empirical probability of mutation $j$ being the descendant of mutation $i$. The probability of a new event with mutation $j$ can be computed as the probability of randomly selecting a node to place the new event multiplied by the entry $w_{ij}$ if the direct ancestor is mutation $i$.
- Frequency: The simplest benchmark is to predict the next mutational events using the relative frequencies of the mutations in the cohort. The probability of a new event with

 

# Article

mutation $j$ can be computed as the probability of randomly selecting a node to place the new event multiplied by the relative frequency of $j$.

For the last two approaches, normalization is required to ensure that the probabilities sum up to 1 over all possible events. We say that a method has better predictive performance if the event that actually happened ranks higher by that method compared to all other possible events. We, therefore, compare the percentile ranks of the next events. In the case of ties, we take the average percentile rank of the tied events.

## Simulations

We use simulation studies to assess the performance of TreeMHN in estimating the network parameters Θ and the probabilities of evolutionary trajectories from a set of mutation trees. Following[30], we first randomly generate a ground truth network Θ with $n$ mutations. The diagonal entries $\log \Theta_{ii}$ are drawn from a uniform distribution ranging from −6 to −1. A random half of the off-diagonal entries $\log \Theta_{ij}$ are set to zero, and another half are sampled from a Gamma distribution $\Gamma(\alpha, \beta)$ with $\alpha = 4$ and $\beta = 2.5$. The non-zero entries are then multiplied by −1 with a 50% chance. These values are chosen to mimic the AML dataset. Given Θ, we then generate $N$ mutation trees (see pseudocode in Supplementary Algorithm 1), from which we can obtain an estimate $\hat{\Theta}$ using TreeMHN. In addition to varying the number of mutations $n$ and the number of trees $N$, we also consider different levels of network sparsity (proportion of zero entries in Θ) and the ratio of positive and negative entries in Θ. Moreover, the tree generating process assumes that each generated tree represents the true mutational history of a tumor. In practice, however, the trees estimated by phylogenetic methods are often noisy. To evaluate the robustness of TreeMHN against the uncertainty in the input tree topologies, we introduce a noise level $\epsilon \in \{1\%, 5\%, 10\%, 20\%\}$, the probability of perturbing individual nodes in the simulated trees, and run TreeMHN on the perturbed trees (Supplementary Section D.3).

We first compute the precision and recall of identifying the edges ($j\,i, j \to i, j\dashv i$) in Θ. Specifically, we call an off-diagonal entry in $\hat{\Theta}$ true positive if and only if it is non-zero and has the correct sign (Supplementary Section D.1). For this metric, we compare TreeMHN against the genotype MHN method, which cannot handle tree structures, since the input is one consensus genotype per tumor. As such, we use the subclonal genotypes as input, weighted by the number of subclones within a tree. Even though the subclonal structure is lost, using multiple genotypes per tumor is still more informative for the genotype MHN method than a consensus genotype.

To benchmark the accuracy of TreeMHN in recovering the trajectory probabilities, we include additionally REVOLVER and HINTRA for comparison. Since these two methods do not have a notion of a sampling event, we use the KL divergence from an estimated trajectory distribution $P_{\hat{\Theta}}$ to the true distribution $P_{\Theta}$ over $\Pi_d$ with $d = 4$ (Eqs. (12) and (13)). Note that longer trajectory lengths can dramatically increase the computational complexity as the number of trajectories in $\Pi_d$ is $\frac{n!}{(n-d)!}$. Since these two alternatives do not output an estimate of Θ, we compute their key matrices (denoted $\boldsymbol{w}$ in REVOLVER and $\beta$ in HINTRA) using the edge frequencies directly from the trees, followed by constructing the probability distributions over $\Pi_d$ with the matrix elements (Supplementary Section D.2).

## Reporting summary

Further information on research design is available in the Nature Portfolio Reporting Summary linked to this article.

## Data availability

The original sequencing data for the AML dataset[21] are available at NCBI BioProject ID [PRJNA648656](), and the associated mutation trees are provided with this paper. The original sequencing data for the NSCLC dataset[15] are available at the European Genome-Phenome Archive under accession code [EGAS00001002247](), and the associated mutation trees are obtained from the R package [evoverse. datasets (v0.1.0)](https://github.com)[67]. The original somatic mutational and clinical data for the breast cancer dataset[32] are available at the cBioPortal for Cancer Genomics with study ID [breast_msk_2018](), and the associated mutation trees[9] are obtained from [https://github.com/elkebir-group/RECAP/tree/master/data/breast_Razavi](). All simulation data, processed mutation trees, and other relevant data are available as Source Data files at Zenodo ([https://doi.org/10.5281/zenodo.7817793]()). Source data are provided with this paper.

## Code availability

The R package for TreeMHN (v0.1.0) and the analysis code are available at both GitHub ([https://github.com/cbg-ethz/TreeMHN]()) and Zenodo ([https://doi.org/10.5281/zenodo.7816776]())[68].

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

## Acknowledgements

The authors would like to thank Katharina Jahn for providing the phylogenetic trees from ref. 21. This work was supported by ETH Zurich [Open ETH project SKINTEGRITY.CH (N.B.)]; the Swiss National Science Foundation [grant number 310030_179518/1 (N.B.)]; and the European Union's Horizon 2020 research and innovation program [grant number 951970, OLISSIPO project (N.B.)].

## Author contributions

Conceptualization, methodology, and writing - review & editing: all authors; Data curation, formal analysis, software, visualization, writing – original draft preparation: X.G.L; Supervision: J.K. and N.B.

## Competing interests

The authors declare no competing interests.
