## [Peer Review File · Nature Communications]

Reviewer comments, first round

Reviewer #1 (Remarks to the Author):

Review of "Joint inference of exclusivity patterns and recurrent trajectories from tumor mutation trees"

This paper introduces TreeMHN, a novel probabilistic framework for inferring recurrent evolutionary trajectories and exclusivity patterns from tumor phylogenetic trees. It has been previously reviewed and presented at RECOMB 2022. While most of the evidence suggests that this is an interesting work and a valuable methodological addition to the existing literature on cancer progression, I have not found it very straightforward to follow the paper, especially the part describing the methods. I noticed that some of the RECOMB reviewers had similar issues. This is perhaps due to the fact that this is still relatively new area of research and research community has quite limited familiarity with this type of methods, including their underlying assumptions, common input formats, terminology, and others. Therefore I strongly recommend the authors to put additional effort into presentation, expand and simplify the methods description and minimize dependence on the existing literature wherever possible. If that may be of some help, I recommend doing something at least similar to as to what they have previously done in their excellently written paper describing SCITE, <https://doi.org/10.1186/s13059-016-0936-x>. Unfortunately I do not find the TreeMHN paper in the current form as clear enough and easy to follow.

I also have some additional questions, suggestions about potential additional experiments and disagreements about some statements made in the manuscript.

Overall, I believe that all of my concerns are addressable and therefore, for this round of review, my recommendation for this submission is a major revision.

COMMENTS and SUGGESTIONS:

1. I found it challenging to understand some important basic details of the method. In particular, I had issues with understanding the following:

- What is the input to this method? I first assumed it to be a set of mutation trees where trees are obtained at the level of individual patients (like in AML dataset). However, after checking Figure 1, I noticed the following sentence "Given an MHN, TreeMHN generates heterogeneous mutation trees (left panel)". Then, in one of the following sentences, one can find "Given a set of trees for a patient cohort (left panel)". So, what is in the left panel in Figure 1? Are the mutation trees shown in this panel given as a part of the input and are static, or they are given as the input but can be updated while the method is run, or something else is the case?
- is there a difference between the terms "trajectory" and "pathway"?
- what is the output of this method? Would it be possible to provide a separate subsection, which would clearly and in a mathematically concise way describe all outputs?
- given a tumor tree, in order to predict the next most likely mutational event, does the method need any additional information other than the parameter values inferred from some cohort of patients? Considering the extent of intra-tumor and inter-tumor heterogeneity, it would come quite surprising that this can be done reliably. Taking into account their size, do the datasets used in this study offer statistically strong evidence that these predictions are accurate and reliable in

practice? I noticed that some applications are presented on Page 6. Have these patients for which predictions were made been excluded from the training? How large the probabilities of the occurrence of a particular event as the next event are? What comes first to my mind is that the method would in most cases report a collection of probable variants with close probabilities, which are usually small numbers. For example, reporting five mutations A,B,C,D,E to be the next most likely events with probabilities 0.05, 0.048, 0.46, 0.044, 0.042 (the list would usually be longer when non-targeted sequencing is employed). How would one exploit this information in patient treatment in real clinical applications?

- In my interpretation of the upper right panel of Figure 1, the method also reports relative time that passes between the occurrence of specific pairs of mutations. Isn't this time very dependent on so many other factors (variants not included in the targeted panels used for generating AML dataset, epigenetics, exogeneous factors etc.) that it is in fact hardly predictable from the currently available DNA sequencing datasets?

- I might later have some additional questions for the other parts of the methods description.

2. Related to Figure 5, CONETT is one of the existing methods for finding repeated evolutionary trajectories. Are there some recurrent trajectories in this dataset that could be identified by CONETT as present in significant number of patients, which are missing in Figure 5? Here, one (not necessarily the best) way to define "significant" is as follows: for each of the 40 trajectories shown in Figure 5, compute the number of patients in which it is present and then take the median of these 40 numbers and set it as a lower threshold in CONETT (i.e., require at least so many patients to have a trajectory in order to report it in the output). In addition, could these 40 numbers be added to Figure 5 as a separate column?

3. I noticed that at several places it is emphasized that this method can handle parallel mutations. One of the sentences related to this is "For computational convenience, most existing tree reconstruction algorithms adopt the infinite sites assumption (ISA), i.e. they require that each mutational event (e.g. an SNV) is gained at most once."

I do not understand why this assumption is problematized and brought up in a tone that suggests it to be a noticeable limitation of tree reconstruction algorithms (maybe I am just misunderstanding its main message). Parallel mutations in the same gene are not challenging for most of such algorithms as soon as they occur at different genomic loci, which is expected to be true in most cases where multiple SNVs affect the same gene. The reason is simple: mutations in the input to these methods are identified by unique mutation IDs, which do not have to be equal to gene names. So, as soon as two SNVs occur at different loci, even if they affect the same gene, they can be assigned unique identifiers distinguishing them. Examples of this can be found in AML trees inferred by SCITE and used in this work. It can occasionally happen that the very same location is mutated twice, but such cases are rare and on top of that the noise that they generate is usually too weak to have a prominent effect on the structure/accuracy of the reconstructed tree. What can be indeed challenging and have an impact on the tree structure are losses of mutations, but they are not handled in this work. Is it perhaps that, instead of referring to tree reconstruction algorithms, the authors were aiming to stress that the methods which are direct competitors to TreeMHN can not handle parallel mutations, whereas TreeMHN can?

4. In the section presenting application to AML dataset it is stated that it was assumed that all point mutations in the same gene have similar effect. Has any distinction been made between synonymous and non-synonymous point mutations or the former are not included at all into the analysis?

Reviewer #2 (Remarks to the Author):

This paper introduces TreeMHN, a method for the joint inference of mutual exclusivity patterns and recurrent trajectories from a cohort of tumor phylogenies. Additionally TreeMHN supports the

prediction of subsequent mutational events given a tree. This is a timely topic, and the results are comprehensive. The paper has been reviewed previously at RECOMB, and the authors were responsive to the feedback. I only have a few comments.

1. Better presentation of results

The results are hard to follow without reading the Methods section.

* While I understand that details need to be delegated to Methods and the supplement, it would help to expand the TreeMHN overview paragraph so that the Results section can be read and understood after reading the Introduction. What precisely does θ_{ij} indicate? Moreover, it would be good to describe the difference between "genotype MHN" and "tree MHN". Moreover, give an informal definition of "trajectory". And discuss the stability selection. Also, provide an informal definition of "evolutionary trajectory".

* Because of the terse TreeMHN overview section, the simulation section in the Results is hard to read. First, no overview of the simulation experiments is given (What range of mutations are you considering? How many patients? How did you run other methods? What is genotype MHN? What is stability selection?). Second, parameter N (the number of trees in the cohort) is not described. Third, the ground truth is not clear -- what precisely is recall and precision showing? Do you ignore the sign of the interaction? Fourth, what are "higher baseline rates", larger number of trees and mutations? Please define baseline rate.

* Figure 3: was stability selection use for TreeMHN and GenotypeMHN?

* AML data, assessment of estimation of conditional probabilities. It is not clear what the TreeMHN baseline method corresponds to. Nor is it clear what the difference is between MHN consensus and MHN weighted. The latter could have been introduced in the TreeMHN overview section. Finally, Figure 6b -- the rationale of comparing against a frequency-based model is not given. One could also use the metric used in 6c, which is more direct.

* Figure 8: convention is to shade the observed variables, not the latent/hidden variables.

2. Assess tree selection

My only other remaining comment is that the TreeMHN model supports computing $p(T | \theta)$ and also supports multiple trees per patient. It would be good to assess using simulations whether the ground-truth tree received maximum probability.

Joint inference of exclusivity patterns and recurrent trajectories from tumor mutation trees

Xiang Ge Luo^{1,2}, Jack Kuipers^{1,2}, Niko Beerenwinkel^{1,2(✉)}

Manuscript # NCOMMS-22-28062

doi.org/10.1101/2021.11.04.467347

Overview

We would like to thank all reviewers for their careful reviews and constructive comments, which have considerably helped with improving the clarity and the quality of the manuscript. We have addressed all comments point by point and incorporated their suggestions into the revision. The major changes are:

- We have put considerable effort into improving the presentation of the results and the readability of the manuscript, including rewriting the entire TreeMHN overview section. We feel that the changes will now allow the reader to more easily understand how TreeMHN works and to follow the Results section without diving into the Methods section.
- We conducted additional experiments, including lowering the number of samples in simulations and comparing TreeMHN to CONETT and MASTRO on the AML trajectories.
- We now elaborate in Methods on all technical details regarding the performance assessment on the predictions of the next mutational events shown in Figure 6. For Figure 6b, we re-ran the experiments by excluding for learning the samples on which the predictions are made.

Other specific changes and additions are provided in the following detailed response (our response in blue, the reviewers' comments in black). All page and reference numbers in our response are based on the revised manuscript, unless otherwise stated. The page and reference numbers mentioned in the reviewers' comments are kept intact and refer to the original manuscript. We sincerely appreciate the time and effort that the reviewers dedicated to reviewing this manuscript.

Reviewer #1

Overall Comments

This paper introduces TreeMHN, a novel probabilistic framework for inferring recurrent evolutionary trajectories and exclusivity patterns from tumor phylogenetic trees. It has been previously reviewed and presented at RECOMB 2022. While most of the evidence suggests that this is an interesting work and a valuable methodological addition to the existing literature on cancer progression, I have not found it very straightforward to follow the paper, especially the part describing the methods. I noticed that some of the RECOMB reviewers had similar issues. This is perhaps due to the fact that this is still relatively new area of research and research community has quite limited familiarity with this type of methods, including their underlying assumptions, common input formats, terminology, and others. Therefore I strongly recommend the authors to put additional effort into presentation, expand and simplify the methods description and minimize dependence on the existing literature wherever possible. If that may be of some help, I recommend doing something at least similar to as to what they have previously done in their excellently written paper describing SCITE, <https://doi.org/10.1186/s13059-016-0936-x>. Unfortunately I do not find the TreeMHN paper in the current form as clear enough and easy to follow.

I also have some additional questions, suggestions about potential additional experiments and disagreements about some statements made in the manuscript.

Overall, I believe that all of my concerns are addressable and therefore, for this round of review, my recommendation for this submission is a major revision.

Comments and suggestions

1. I found it challenging to understand some important basic details of the method. In particular, I had issues with understanding the following:
 - What is the input to this method? I first assumed it to be a set of mutation trees where trees are obtained at the level of individual patients (like in AML dataset). However, after checking Figure 1, I noticed the following sentence "Given an MHN, TreeMHN generates heterogeneous mutation trees (left panel)". Then, in one of the following sentences, one can find "Given a set of trees for a patient cohort (left panel)". So, what is in the left panel in Figure 1? Are the mutation trees shown in this panel given as a part of the input and are static, or they are given as the input but can be updated while the method is run, or something else is the case?

The input of the method is indeed a set of mutation trees obtained at the level of individual patients, which is static and not updated while the method is run. The tree-generating process is the modeling assumption of TreeMHN; it should not be confused with the inference procedure which solves the inverse problem, i.e., given a set of trees to find the MHN. To avoid confusion, we have removed the “generate” arrow from Figure 1, which now represents the inference process, and clarified in the TreeMHN overview section what the input is:

p.3 *“To perform inference with TreeMHN, the input is a set of N independent tumor mutation trees with a total number of n mutations, which may be constructed from bulk, multi-region, or single-cell sequencing data using phylogenetic methods...”*

- Is there a difference between the terms "trajectory" and "pathway"?

No, we used the terms “evolutionary trajectories” and “mutational pathways” interchangeably in the previous manuscript. Both terms refer to sequences of mutations and can be found in existing literature. We agree that this is confusing for the reader. Therefore, we now use only “trajectories” or “evolutionary trajectories” in the manuscript.

- What is the output of this method? Would it be possible to provide a separate subsection, which would clearly and in a mathematically concise way describe all outputs?

Given a set of N mutation trees with a total number of n mutations as input, the output of TreeMHN is an n -by- n matrix, representing the Mutual Hazard Network. The diagonal entries of this matrix indicate how often each mutation will occur and fixate independent of the other mutations. The off-diagonal entries encode the exclusivity and co-occurrence patterns of mutations. Conditioned on the estimated matrix, we can compute the probabilities of different evolutionary trajectories or evaluate the most likely next mutational events given a tumor tree. While keeping the mathematical details in Methods, we have revised the TreeMHN overview section considerably, which should now clarify conceptually what the output of TreeMHN is:

p.3 *“... The output is $\hat{\Theta}$, an estimated MHN describing the exclusivity and co-occurrence patterns of mutations for the given cohort [...] Furthermore, the estimated model allows us to compute the probabilities of different evolutionary trajectories or evaluate the most likely next mutational events given a tumor tree using the properties of exponential distributions ...”*

- Given a tumor tree, in order to predict the next most likely mutational event, does the method need any additional information other than the parameter values inferred from some cohort of patients? Considering the extent of intra-tumor and inter-tumor heterogeneity, it would come quite surprising that this can be done reliably. Taking into account their size, do the datasets used in this study offer statistically strong evidence that these predictions are accurate and reliable in practice? I noticed that some applications are presented on Page 6. Have these patients for which predictions were made been excluded from the training? How large the probabilities of the occurrence of a particular event as the next event are? What comes first to my mind is that the method would in most cases report a collection of probable variants with close probabilities, which are usually small numbers. For example, reporting five mutations A,B,C,D,E to be the next most likely events with probabilities 0.05, 0.048, 0.46, 0.044, 0.042 (the list would usually be longer when non-targeted sequencing is employed). How would one exploit this information in patient treatment in real clinical applications?

- TreeMHN only requires the estimated Mutual Hazard Network to predict the most likely next mutational events given a tumor tree. No other information is needed, as stated in the Methods (Eq (14)) and illustrated in Figure 1.
- The extensive intra-tumor and inter-tumor heterogeneity indeed makes predicting the next event challenging; hence the probability of any particular event is low. Even though it is a challenging task, we still wish to make the best predictions we can, which is why we make our best efforts to evaluate how reliable the TreeMHN predictions are. We would like to emphasize that TreeMHN is the first and only method we know of to compute the probability of an event given a tumor tree. Nevertheless, we adapt five alternative approaches (now elaborated in Methods) to this task and show that TreeMHN predictions are statistically significantly better (Figure 6b).
- In the previous version of the manuscript, we did exclude the longitudinal samples from training for all methods but missed that for the retrospective case (for all methods). Thanks to the reviewer’s comment, this is now corrected and clarified:

p.14 *“... For retrospective predictions, we first take a primary tumor tree and exclude it from training. Then, we enumerate all its rooted subtrees and their corresponding downstream events, followed by computing the probability of each downstream event conditioned on its parent subtree and the estimated network (Eq. (14)). For forward predictions, on the other hand, we take the longitudinal samples and consider only those tree pairs on consecutive time points, where the second tree has mutations not present in the first tree. We compute the probabilities for the new events using the matrix estimated from the 123 primary tumor trees (i.e. longitudinal samples are excluded from training) ...”*

- We now display the probabilities of the next events in Supplementary Figure S17 & S18 for both retrospective predictions on rooted subtrees and forward predictions on longitudinal samples. In Figure S17, the probabilities are seemingly small (mostly between 0.01 and 0.1) because we only focus on the events that actually happened. In Figure S18, we display the probability distributions of all possible events given a tree and observe that not all events receive similar probabilities. For TreeMHN in particular, most of the events have very low probabilities, and only a few of them, including seven out of the nine new events reported in the main text, are orders of magnitude higher. When there are many possible events, it is natural that their probabilities are small. Therefore, we are interested rather in their relative probabilities. As noted in the revised manuscript,

p.14 *“... We say that a method has better predictive performance if the event that actually happened ranks higher by that method compared to all other possible events. We therefore compare the percentile ranks of the next events. In the case of ties, we take the average percentile rank of the tied events.”*

- We agree that evolutionary prediction is in general a very difficult task. Related to the next question, reliable predictions also depend on many other factors, such as unknown driver mutations, epigenetics, tumor microenvironmental changes, etc. To bring such results to clinical

applications, we still need real experiments to verify the events that receive higher rankings and further develop the model. Nevertheless, we find it very encouraging that TreeMHN shows improved results on this task and opens up more opportunities for precision medicine.

- In my interpretation of the upper right panel of Figure 1, the method also reports relative time that passes between the occurrence of specific pairs of mutations. Isn't this time very dependent on so many other factors (variants not included in the targeted panels used for generating AML dataset, epigenetics, exogenous factors etc.) that it is in fact hardly predictable from the currently available DNA sequencing datasets?

Since we model the waiting times until the occurrence of new subclones using exponential distributions, it is mathematically straightforward to compute the probabilities and the expected waiting times of evolutionary trajectories as well as the probabilities of the next mutational events given a tumor tree using the properties of competing exponentials (the minimum of a set of independent exponential random variables). However, like all other cancer progression models, we assume that we have included all important mutational events and the tumor microenvironment stays unchanged over time. Even without hidden factors, the expected times are relative to the sampling rate, which is assumed to be 1 for identifiability, so the reported times cannot be interpreted as real time. We nevertheless show the expected waiting times in the manuscript to demonstrate that in principle this can be done by TreeMHN (and waiting times may be interpretable with additional knowledge about the sampling time relative to the start of the evolutionary process), whereas alternative methods such as REVOLVER generally can not provide such estimates. With increasing longitudinal and clinical data in the future, it may be possible to extend TreeMHN and estimate the times more accurately. We now talk about this in the Discussion:

p.10 *“... Given the estimated parameters, TreeMHN allows us to compute the probabilities of different evolutionary trajectories and the expected waiting times between mutational events. However, in general, these waiting times cannot be interpreted as real calendar time since they are with respect to the unknown sampling times and the scaling factor is therefore unknown. One remedy is to use longitudinal data, where the sampling time is either provided or can be inferred from data. Including observed sampling times is technically possible [54], but such data are often difficult to obtain without having treatment interventions. Modelling drug response data, or including other exogenous factors, is a crucial but challenging direction to explore, for which TreeMHN may serve as a basis.”*

- I might later have some additional questions for the other parts of the methods description.
2. Related to Figure 5, CONETT is one of the existing methods for finding repeated evolutionary trajectories. Are there some recurrent trajectories in this dataset that could be identified by CONETT as present in significant number of patients, which are missing in Figure 5? Here, one (not necessarily the best) way to define "significant" is as follows: for each of the 40 trajectories shown in Figure 5, compute the number of patients in which it is present and then take the median of these 40 numbers and set it as a lower threshold in CONETT (i.e., require at least so many

patients to have a trajectory in order to report it in the output). In addition, could these 40 numbers be added to Figure 5 as a separate column?

As suggested by the reviewer, we now include in Supplementary Figure S12 a comparison with CONETT on the top 40 most probable evolutionary trajectories inferred by TreeMHN from the AML dataset. Specifically, we add a new column indicating the number of times each trajectory appears in the transitive closures of the mutation trees, and the median of the 40 numbers is 10.5. The germline-rooted evolutionary trajectory tree computed by CONETT, which is conserved in at least 11 patients at 10% significance level, is highly consistent with the TreeMHN most probable trajectories. In particular, none of the trajectories identified by CONETT are missing in the TreeMHN analysis.

Moreover, MASTRO (Pellegrina & Vandin, 2022) is a newly published approach that extends the idea of CONETT by searching for all possible trajectories conserved in at least a certain number of patients, instead of a single conserved evolutionary trajectory tree that can describe the pattern in as many tumors as possible. In Supplementary Figure S13 we show that the majority of the most significant trajectories reported by MASTRO (Figure S5 of Pellegrina & Vandin, 2022) can be found in the most probable trajectories inferred by TreeMHN. Therefore, the TreeMHN results are more comprehensive given the additional estimates of trajectory probabilities, expected waiting times, and probabilities of future events given a tumor tree.

To avoid adding too much technical information in the main text and facilitate readability for broader audiences, we keep both figures in the supplementary materials and add cross-references in Results:

“Next, we infer the most probable evolutionary trajectories from the estimated network (Methods & Figure 5), which are consistent with the significantly conserved ones identified by CONETT and MASTRO (Supplementary Figures S11 & S12).”

3. I noticed that at several places it is emphasized that this method can handle parallel mutations. One of the sentences related to this is "For computational convenience, most existing tree reconstruction algorithms adopt the infinite sites assumption (ISA), i.e. they require that each mutational event (e.g. an SNV) is gained at most once." I do not understand why this assumption is problematized and brought up in a tone that suggests it to be a noticeable limitation of tree reconstruction algorithms (maybe I am just misunderstanding its main message). Parallel mutations in the same gene are not challenging for most of such algorithms as soon as they occur at different genomic loci, which is expected to be true in most cases where multiple SNVs affect the same gene. The reason is simple: mutations in the input to these methods are identified by unique mutation IDs, which do not have to be equal to gene names. So, as soon as two SNVs occur at different loci, even if they affect the same gene, they can be assigned unique identifiers distinguishing them. Examples of this can be found in AML trees inferred by SCITE and used in this work. It can occasionally happen that the very same location is mutated twice, but such cases are rare and on top of that the noise that they generate is usually too weak to have a prominent effect on the structure/accuracy of the reconstructed tree. What can be indeed challenging and have an impact on the tree structure are losses of mutations, but they are not handled in this work. Is it perhaps that, instead of referring to tree reconstruction algorithms, the authors were aiming to

stress that the methods which are direct competitors to TreeMHN can not handle parallel mutations, whereas TreeMHN can?

Yes, we wanted to emphasize that TreeMHN can handle parallel mutations, while many of the alternative approaches cannot, such as REVOLVER, HINTRA, and RECAP. In fact, CONETT and MASTRO can also handle parallel mutations, but these two approaches do not provide probabilistic measures of future events given a trajectory or a tree and thus cannot be easily adapted for evolutionary predictions. The reviewer is correct that this is not a major drawback of the tree reconstruction algorithms. We have revised the manuscript as follows:

p.11 “... For computational convenience, many of the alternatives capable of inferring repeated evolutionary trajectories (e.g., REVOLVER, HINTRA, RECAP) require the infinite sites assumption (ISA), i.e. each mutational event (e.g., an SNV) is gained at most once ...”

Currently TreeMHN does not take into account losses of mutations, and we have noted this limitation in Discussion:

p.10 “... Like all other progression models, however, TreeMHN currently does not consider back mutations, i.e. situations in which a mutation is acquired at first but subsequently lost [57, 58]. A possible extension along this line is to include additional parameters and use as input phylogenetic trees inferred by methods such as SCARLET [59], which views a decrease in copy numbers that overlap a mutated locus as evidence of back mutations ...”

4. In the section presenting application to AML dataset it is stated that it was assumed that all point mutations in the same gene have similar effect. Has any distinction been made between synonymous and non-synonymous point mutations or the former are not included at all into the analysis?

The AML dataset we analyzed does not include any synonymous mutations (see Supplementary Data 1 of Morita et al., 2020). We now add this information to the manuscript:

p.5 “We apply TreeMHN to the cohort of $N = 123$ AML patient samples analyzed in [21] by high-throughput single-cell panel sequencing, which involves 543 somatic mutations in $n = 31$ cancer-associated genes and does not include any synonymous mutations ...”

Reviewer #2

Overall Comments

This paper introduces TreeMHN, a method for the joint inference of mutual exclusivity patterns and recurrent trajectories from a cohort of tumor phylogenies. Additionally TreeMHN supports the prediction of subsequent mutational events given a tree. This is a timely topic, and the results are comprehensive. The paper has been reviewed previously at RECOMB, and the authors were responsive to the feedback. I only have a few comments.

Comments and suggestions

1. Better presentation of results

The results are hard to follow without reading the Methods section.

- While I understand that details need to be delegated to Methods and the supplement, it would help to expand the TreeMHN overview paragraph so that the Results section can be read and understood after reading the Introduction. What precisely does θ_{ij} indicate? Moreover, it would be good to describe the difference between "genotype MHN" and "tree MHN". Moreover, give an informal definition of "trajectory". And discuss the stability selection. Also, provide an informal definition of "evolutionary trajectory".

We have revised the Introduction and the entire TreeMHN overview section by summarizing conceptually and comprehensively the technical details from Methods to enable a smooth transition from Introduction to Results:

- We explain in words the meaning of θ_{ij} and modify the middle panel of Figure 1 to help the readers to visualize the equivalence between the matrix off-diagonal entries and the edges in the network:

p.3 *“... Mutations can have positive (co-occurring), negative (exclusive), or zero (no) effects on the rates of downstream mutations. This is encoded by the off-diagonal entries $\{\theta_{ij}\}_{i,j \in [n]}$ with an equivalent graphical structure (Figure 1). If θ_{ij} is positive, mutation j increases the rate of mutation i (denoted by an edge $j \rightarrow i$). If θ_{ij} is negative, mutation j decreases the rate of mutation i (denoted $j \rightarrow i$)...”*

- In the Introduction, we first bring up the genotype MHN method in the summary of cancer progression models (CPMs):

p.2 *“... A recent method, called Mutual Hazard Networks (MHNs), does not explicitly group exclusive mutations but re-parameterizes mutational waiting times using a matrix that encodes both co-occurrence and exclusivity ...”*

Later, we point out the main difference between TreeMHN and CPMs, including the genotype MHN:

p.2 *“... Unlike classical CPMs, including the genotype MHN method of Schill et al., TreeMHN considers the complete mutational histories of tumors and their subclonal structures represented by the intra-tumor mutation tree rather than the overall presence and absence of mutations ...”*

→ We provide an informal definition of “evolutionary trajectory”:

p.2 - p.3 *“... Each path from the root to a node in the tree constitutes an evolutionary trajectory π , which characterizes the successive accumulation of mutations and uniquely defines a subclone ...”*

→ We discuss stability selection before simulation results:

p.3 *“... The number of parameters to estimate (n^2) often exceeds the number of observations (N). To prevent model overfitting, we can run TreeMHN with stability selection [33], where the parameters in Θ are estimated over many subsamples of the trees, and only those having a high probability of being non-zero are kept. This procedure can significantly improve the precision of identifying the true relationships among the mutations ...”*

- Because of the terse TreeMHN overview section, the simulation section in the Results is hard to read. First, no overview of the simulation experiments is given (What range of mutations are you considering? How many patients? How did you run other methods? What is genotype MHN? What is stability selection?). Second, parameter N (the number of trees in the cohort) is not described. Third, the ground truth is not clear -- what precisely is recall and precision showing? Do you ignore the sign of the interaction? Fourth, what are "higher baseline rates", larger number of trees and mutations? Please define baseline rate.

Along with rewriting the TreeMHN overview section, as suggested by the reviewer, we have also made considerable revision to the simulation section in Results. In particular,

→ We now provide an overview of the simulation experiments:

p.4 *“Through simulations, we assess the performance of TreeMHN in comparison to alternative methods in estimating exclusivity patterns and associated distributions of evolutionary trajectories from mutation trees. For each simulation run, we randomly generate a ground truth network Θ with n mutations and a set of N mutation trees. Using the N simulated trees as input, we run TreeMHN along with the genotype MHN method [30], REVOLVER [7], and HINTRA [8]. We consider different configurations of simulation experiments, including varying the number of mutations n and the number of trees N . For each configuration, we perform 100 repetitions. More simulation details are provided in the Methods.”*

→ Regarding the ground truth network, precision, and recall, we provide a brief description in the simulation results as well as a formal definition in Supplementary Section D.1:

p.4 *“We first evaluate how well TreeMHN can estimate the patterns of clonal co-occurrence and exclusivity by computing the structural differences between the estimated network $\hat{\Theta}$ and the ground-truth network Θ . Specifically, we measure the precision and recall of identifying the true edges in Θ . An estimated off-diagonal entry in $\hat{\Theta}$ is a true positive if and only if it is non-zero and has the correct sign (Supplementary Figure S3) ...”*

In both places, we have made it clear that we take into account the signs of the interactions when assessing the structural differences.

→ We define the baseline rates in the TreeMHN overview, which is before the simulation overview:

p.3 *“... The diagonal entries $\{\Theta_{ij}\}_{i,j \in [n]}$ are the baseline rates of evolution, indicating how quickly each mutation will occur and fixate in a subclone independent of the other mutations ...”*

→ Other details, including how we run alternative methods, are provided in the Methods section and Supplementary Section D in order to keep simulation results concise.

- Figure 3: was stability selection use for TreeMHN and GenotypeMHN?

Yes, we used stability selection for both TreeMHN and the genotype MHN, and we now clarify this in simulation results:

p.5 *“... For TreeMHN and genotype MHN, we additionally implemented the stability selection procedure over a set of regularization parameters $\gamma \in \{0.05, 0.1, 0.5, 1, 1.5, 2, 2.5, 3\}$, where larger values of γ promote sparser network structures and prevent model overfitting ...”*

However, it is computationally too demanding to run genotype MHN with stability selection for $n = 20$. Hence, we omit the corresponding curves in Figure 3 given the clear gaps between the two methods in other cases. This information is provided in the figure caption.

- AML data, assessment of estimation of conditional probabilities. It is not clear what the TreeMHN baseline method corresponds to. Nor is it clear what the difference is between MHN consensus and MHN weighted. The latter could have been introduced in the TreeMHN overview section. Finally, Figure 6b -- the rationale of comparing against a frequency-based model is not given. One could also use the metric used in 6c, which is more direct.

We now provide all technical details, including the description of alternative methods, in the Methods section:

p.14 “... We evaluate TreeMHN predictions on the AML dataset [21]. We perform both retrospective predictions on the rooted subtrees of the 123 primary tumor trees and forward predictions using the longitudinal samples from 15 patients in the same dataset (Figure 6a) [...] While TreeMHN is unique in this task, we adapt five alternative methods for comparison ...”

As suggested by the reviewer, we have replaced the metric in Figure 6b with the metric in Figure 6c, which is the percentile rank:

p.14 “... We say that a method has better predictive performance if the event that actually happened ranks higher by that method compared to all other possible events. We therefore compare the percentile ranks of the next events. In the case of ties, we take the average percentile rank of the tied events. ”

- Figure 8: convention is to shade the observed variables, not the latent/hidden variables.

We have modified Figure 8 such that only the observed variable, the mutation tree structure, is shaded.

2. Assess tree selection

My only other remaining comment is that the TreeMHN model supports computing $p(T | \Theta)$ and also supports multiple trees per patient. It would be good to assess using simulations whether the ground-truth tree received maximum probability.

While TreeMHN supports computing $p(T | \Theta)$, it is not a phylogenetic tree reconstruction algorithm. Instead of taking patient-level sequencing data as input to reconstruct a phylogenetic tree, we take a collection of trees across patients obtained from independent phylogenetic methods (e.g. SCITE) to infer the underlying Θ . In other words, we cannot assess whether the ground-truth tree receives the maximum probability, because this probability is with respect to the sequencing data which we do not work with directly. Apart from being computationally much more efficient, working directly with the trees allows for greater flexibility of adapting TreeMHN to future improvements in phylogenetic methods. It is also possible to combine different phylogenies from various sources (e.g. SCITE, SiFit, SCIΦN, etc.) to take into account different modeling assumptions. We are aware that input tree structures can be noisy and thereby evaluated the robustness of TreeMHN against such uncertainty in simulations (see Supplementary Section D.3, Supplementary Figure S5 & S6). For real data applications, one may sample the trees for each patient proportionally to their posterior probabilities and take them as weighted inputs to TreeMHN.

Reviewer #3

Overall Comments

Overall, a nice conceptual idea (and probabilistic modeling work) that clearly extends current theories and tools. While conceptually superior, I am not entirely convinced that the results are markedly better on the real datasets, which are often quite sparse/heterogeneous.

Comments and suggestions

1. Need to better describe the nature of the study cohort. Does TreeMHN require trees deriving from independent tumors? If the tumors are not independent, i.e., in the settings of multi-region/longitudinal sampling of same patients, how does TreeMHN deal with the interdependence, and would the accuracy be affected?

Yes, TreeMHN takes trees reconstructed from independent tumors as input. These trees can be derived from single-cell, multi-region (sampled from the same tumor), or bulk sequencing data. This is now clarified in the revised TreeMHN overview section:

p.3 *“To perform inference with TreeMHN, the input is a set of N independent tumor mutation trees with a total number of n mutations, which can be constructed from bulk, multi-region, or single-cell sequencing data using phylogenetic methods...”*

While the current version of TreeMHN does not account for the case of metastasis, where tumors are in different locations of the same body, or the case of longitudinal sampling, we believe it can serve as a basis for future methods.

2. While illustrating the points, simulation parameters ($n < 100$, $N > 100$) do not match those from a typical study cohort ($n > 100$, $N < 200$), e.g., the AML cohort, which often have more mutations and less patients. In what ranges of parameters that TreeMHN perform well, comparatively to other simpler (yet likely more robust) programs?

Yes, the reviewer is correct that the real datasets often have more mutations and fewer patients. By adding the case $N = 100$ in simulation experiments, we see that the performance continues to drop with decreasing sample size. In general, the situation when the number of parameters to estimate is much larger than the number of samples available, i.e., the $N \ll p$ setting (Bühlmann & van de Geer, 2011), poses difficulty in robust parameter estimation and is not unique to TreeMHN. We are aware of this problem and hence implement the stability selection procedure (Meinshausen & Bühlmann, 2010), where the parameters are estimated over many subsamples of the trees, and only those having a high probability of being non-zero are kept (Supplementary Section B). We show in simulations that stability selection effectively improves the precision of identifying the true parameters although at the cost of lower recall. In real data analyses, we continue to apply stability selection to reduce the number of false positives.

Apart from stability selection, one may need additional restrictions on the set of mutational events being considered in order to gain enough statistical power to detect any interactions between events. For

example, the AML dataset involves 123 patients and 543 point mutations in 31 genes (Morita et al., 2020). At the point mutation level, the same event almost never recurs across the patients, making it impossible for any methods to infer statistically significant signals between events given the small sample size. To increase statistical power, when collecting more data cannot be done in the short run (see related question below), we choose to summarize the events at the gene level by assuming all non-synonymous point mutations in the same gene to have similar effects. This assumption is also biologically motivated and not uncommon across state-of-the-art methods for detecting repeated evolutionary trajectories (REVOLVER, HINTRA, RECAP, CONETT, MASTRO) from a small set of heterogeneous mutation trees. Even after reducing the number of events, interactions between rare events are still hard to detect, and the corresponding parameters will be set to zero by stability selection (see Supplementary Figure S11 & S19). Hence, there is a tradeoff between considering each unique event versus a smaller set of more coarse-grained events, which allows for more efficient inference of their dependencies..

In both simulations and real data applications, we have benchmarked TreeMHN against both simple baselines and the state-of-the-art methods that we know of, including adding new comparisons with CONETT and MASTRO for the AML dataset as suggested by Reviewer 1 (Supplementary Figure S12 & S13). The TreeMHN results are consistent with other methods and demonstrate superior performance.

3. While overall writing (including Methods/Sup. Doc) is fine, many detailed concepts/approaches appear cryptic to readers in the main text. Please streamline.
 - a. Page 3, it will be helpful to briefly describe how simulation was done and why is it realistic, before presenting the evaluation results. Current presentation is technical/superficial, making it difficult to appreciate the biological relevance/significance of the results.

As suggested by the reviewer, we now provide an overview of the simulation experiments:

p.4 *“Through simulations, we assess the performance of TreeMHN in comparison to alternative methods in estimating exclusivity patterns and associated distributions of evolutionary trajectories from mutation trees. For each simulation run, we randomly generate a ground truth network Θ with n mutations and a set of N mutation trees. Using the N simulated trees as input, we run TreeMHN along with the genotype MHN method [30], REVOLVER [7], and HINTRA [8]. We consider different configurations of simulation experiments, including varying the number of mutations n and the number of trees N . For each configuration, we perform 100 repetitions. More simulation details are provided in the Methods.”*

- b. Page 4, “At the same regularization level, the solutions with stability selection achieve much higher precision although at the cost of lower recall.” At this point, the readers do not know what is “stability selection”.

Yes, we agree that this is confusing. We now introduce stability selection as well as regularization before showing the results:

p.3 “... The number of parameters to estimate (n^2) often exceeds the number of observations (N). To prevent model overfitting, we can run TreeMHN with stability selection [33], where the parameters in Θ are estimated over many subsamples of the trees, and only those having a high probability of being non-zero are kept. This procedure can significantly improve the precision of identifying the true relationships among the mutations ...”

p.5 “... For TreeMHN and genotype MHN, we additionally implemented the stability selection procedure over a set of regularization parameters $\gamma \in \{0.05, 0.1, 0.5, 1, 1.5, 2, 2.5, 3\}$, where larger values of γ promote sparser network structures and prevent model overfitting ...”

- c. “We also notice that both methods perform better in recovering the pairwise interactions between mutations with higher baseline rates than those with lower baseline rates (Figure 2, Supplementary Figure S4)”. Neither figure has any indication of mutation rates. The authors should avoid presenting too much technical details/concepts in the main figures/text, without being able to clearly define them a priori.

Yes, as also pointed out by Reviewer 2, this confusion is caused by missing relevant information before showing the simulation results. In the revised manuscript, we first define the baseline rates in TreeMHN overview:

p.3 “... The diagonal entries $\{\theta_{ij}\}_{i,j \in [n]}$ are the baseline rates of evolution, indicating how quickly each mutation will occur and fixate in a subclone independent of the other mutations ...”

Then, we discuss the equivalence between the off-diagonal entries of the matrix Θ and the edges in the network, along with a clear illustration in the middle panel of Figure 1:

p.3 “... Mutations can have positive (co-occurring), negative (exclusive), or zero (no) effects on the rates of downstream mutations. This is encoded by the off-diagonal entries $\{\theta_{ij}\}_{i,j \in [n]}$ with an equivalent graphical structure (Figure 1). If θ_{ij} is positive, mutation j increases the rate of mutation i (denoted by an edge $j \rightarrow i$). If θ_{ij} is negative, mutation j decreases the rate of mutation i (denoted $j \rightarrow i$)...”

The first part of the simulation experiments is to compare the structural differences between an estimated network and the ground truth network by counting the number of correctly identified edges. We now make this point clear before showing the results:

p.4 “We first evaluate how well TreeMHN can estimate the patterns of clonal co-occurrence and exclusivity by computing the structural differences between the estimated network $\hat{\Theta}$ and the ground-truth network Θ . Specifically, we measure the precision and recall of identifying the true edges in Θ . An estimated off-diagonal entry in $\hat{\Theta}$ is a true positive if and only if it is non-zero and has the correct sign (Supplementary Figure S3) ...”

After supplying the above information, we believe that the readers should be able to understand Figure 2 and Supplementary Figure S4. The difference is that the former includes the top half of the mutations with higher baseline rates, whereas the latter includes all mutations. The goal here is to illustrate that the interactions between recurrent events can be recovered more reliably than rare events. This observation and the associated reasoning are discussed in text:

p.5 *“... By sorting the rows and columns of $\hat{\Theta}$ by descending baseline rates [...] The reason is that interactions between rare mutations can hardly be observed in the trees, leading to a lack of statistical power to correctly estimate the corresponding entries in Θ . This limitation is in general method-independent.”*

- d. “Across different sample and network sizes”. What does “network” mean? Authors should try to use consistent terminologies. Another example is that “trajectory” and “pathway” were used interchangeably, which could be misleading to readers.

Yes, the terminologies should be more consistent throughout the manuscript. To avoid confusion, we replace “network size” by “the number of mutations” and use only the term “trajectory” after revision.

4. What order (rows or columns?) should the readers follow to read the MHN plots, e.g., Fig. 4? In other words, which axis corresponds to i (ancestors), and which to j (descendants)? Directionality is unclear. Would it be helpful to also label genes on the X axis?

We agree that the directionality can be confusing for the readers. Following the paper that first introduced the concept of Mutual Hazard Networks (Schill et al., 2020), a non-zero off-diagonal entry θ_{ij} corresponds to an edge from j (ancestors) to i (descendants), which is now illustrated in the middle panel of Figure 1. To make this clear throughout the paper, we label the columns of the MHN plots with “ancestor” and rows with “descendant”.

5. The AML study was based only on data from the Nature Communication paper. It will be great if the authors can try to consolidate data/observations from the Nature (Ross Levine) paper as well (which was published at similar time). For example, in the Nature paper, FLT3 and NRAS mutations are often late instead of early events. And that NPM1 and NRAS mutations rarely cooperate. Were these observations contradictory to the pathways shown in Fig. 5? Why do the first mutations appear at similar timing in Fig. 5? These need to be clarified if not improved.

→ Since the start of the project in 2020, we have been making sustained efforts to consolidate the AML datasets from various sources, which involves three research groups from different institutes. As of now this task is still ongoing. We will continue to push this forward, but the consolidated dataset is still being prepared, and this effort is beyond the scope of the present manuscript.

→ Regarding Figure 5, we do not find the reported trajectories contradictory to the results in the Nature paper. First, *FLT3* and *NRAS* mutations are not always late events. In a number of cases they can also serve as initiating mutations or exist in single-mutant clones, as illustrated in Fig. 3c and Extended Data Fig. 3e of the Nature paper. In the AML dataset we analyzed, 18 out of the 123 tumor trees have *FLT3* or *NRAS* as early events (see figure below for a few examples). This ratio is not over-represented in Figure 5, where 6 out of the top 40 trajectories have one of these two mutations as the first event, and these trajectories can all be found in the observed trees (see the new Supplementary Figure S12). Moreover, the estimated network in Figure 4 shows that *DNMT3A* and *IDH2* mutations have the highest baseline rates and positive effects on the occurrence of many other mutations (e.g. *RUNX1*, *SRSF2*), indicating that they are more often early events and promoting the occurrence of downstream mutations, which is consistent with the Nature paper.

Second, the Nature paper states that the RAS-*NPM1* co-mutant clones are significantly larger than the RAS mutant-only clones but not significantly larger than the single-mutant *NPM1* clones. They did not say that RAS and *NPM1* mutations are mutually exclusive. In fact, they even show a trajectory involving *NPM1* and *NRAS* in their Fig. 3b. Therefore, it is not contradictory to have *NPM1* and *NRAS* co-occurring in the most probable evolutionary trajectories. The size of the co-mutant clone, is not information the TreeMHN model can infer, and we have noted this limitation in Discussion:

p.10 *“Nevertheless, TreeMHN does not take into account the subclone sizes, which can be viewed as consequences of clonal selection [62]. As noted by [63], inferring mutation rates and fitness values jointly may have promise but is challenging. On the one hand, larger subclone sizes can be attributed to both their earlier appearance or higher fitness [63]. On the other hand, the mutation rates in cancer progression models are rates of evolution, which implicitly involve subclone fitness. Thus, any attempts to modify TreeMHN to model clonal selection need to be taken with caution ...”*

Due to the small sample size, each paper may find slightly different patterns given the extensive heterogeneity in the tumor samples, but the overall conclusions should be consistent. We hope to gain more insight into AML after integrating the datasets in the future, and methods like the TreeMHN we developed here will be crucial for such analyses.

- The last question relates to the expected waiting times between events. In TreeMHN, we model the waiting times until the occurrence of the next mutations using independent exponential random variables. Suppose there are three mutations A, B, and C, of which the waiting times are independent and exponentially distributed with rates λ_A , λ_B , and λ_C . By competing exponentials, the minimum waiting time of the three mutations is also exponentially distributed with rate $\lambda_A + \lambda_B + \lambda_C$, and the expected waiting time until one of these three mutations occurs given that it is the first event is $\frac{1}{\lambda_A + \lambda_B + \lambda_C}$. In other words, no matter which mutation comes first, mathematically the expected waiting time is the same. Therefore, the first mutations in Fig. 5 share the same expected waiting time. We talk about the expected waiting times in Discussions:

p.10 *“... Given the estimated parameters, TreeMHN allows us to compute the probabilities of different evolutionary trajectories and the expected waiting times between mutational events. However, in general, these waiting times cannot be interpreted as real calendar time since they are with respect to the unknown sampling times and the scaling factor is therefore unknown. One remedy is to use longitudinal data, where the sampling time is either provided or can be inferred from data. Including observed sampling times is technically possible [54], but such data are often difficult to obtain without having treatment interventions. Modelling drug response data, or including other exogenous factors, is a crucial but challenging direction to explore, for which TreeMHN may serve as a basis.”*

6. Fig. 6 legend is in sort of paragraph style, not figure legend style, and is inconsistent with the legend styles in other figures. I also have difficulty understand what Fig. 6b means? What are methods being compared: e.g., TreeMHN (baseline), MHN (weighted) ...? These were not defined in the context. What is the last column “Frequency” in Fig. 6c, “frequency based model”? Again, perhaps too much technical details to explain in the main text and for the readers to follow. In Fig. 6c, did I understand correctly that there is little performance gain over RESOLVER: 91.7 vs 89.2 on this cohort? Why is that? Any explanation? [REVOLVER is worse in the retrospective case]

- In the previous manuscript, we put too much technical information in the legend of Figure 6, making it unnecessarily long and confusing. We now elaborate on the related technical details in the Methods section, including the description of alternative approaches being compared, which were originally in the supplementary materials:

p.14 “... We evaluate TreeMHN predictions on the AML dataset [21]. We perform both retrospective predictions on the rooted subtrees of the 123 primary tumor trees and forward predictions using the longitudinal samples from 15 patients in the same dataset (Figure 6a) [...] While TreeMHN is unique in this task, we adapt five alternative methods for comparison ...”

To improve interpretability of the plots, as suggested by Reviewer 2, we have replaced the metric in Figure 6b with the metric in Figure 6c, which is the percentile rank:

p.14 “... We say that a method has better predictive performance if the event that actually happened ranks higher by that method compared to all other possible events. We therefore compare the percentile ranks of the next events. In the case of ties, we take the average percentile rank of the tied events. ”

We also report the estimated probabilities of the next events in Supplementary Figure S17 & S18 for both retrospective and forward predictions. As discussed in our reply to Reviewer 1 on page 4, the rationale of reporting the percentile ranks instead of the probabilities is that it is only meaningful to discuss how large the probabilities of the next events are if we know the probabilities of the other possible events.

- We investigated closely into why the predictive performance of REVOLVER on longitudinal samples is better than in the retrospective case. Other than limited sample size, we found that the precise definition of percentile ranks on tie cases can largely change the results. Note that REVOLVER is not originally designed for predictions. We describe in Methods how we adapt it for such tasks:

p.14 “REVOLVER: We first convert the subclonal genotypes into cancer cell fractions, the proportion of cancer cells having a specific mutation, followed by running REVOLVER to obtain the information transfer matrix w (Supplementary Figure S16). An entry w_{ij} is the number of times mutation i occurs before mutation j in the patients. By row-normalizing w , we can obtain the empirical probability of mutation j being the descendant of mutation i . The probability of a new event with mutation j can be computed as the probability of randomly selecting a node to place the new event multiplied by the entry w_{ij} if the direct ancestor is mutation i .”

We add a new Supplementary Figure S16 to show the normalized information transfer matrix learned by REVOLVER on the 123 primary tumor samples. Using this sparse matrix to compute the probabilities of next events can result in many ties, as illustrated by the probability distributions in Supplementary Figure S18. In particular, we observe that the event “Root → *SF3B1* → *SRSF2* → *NRAS* → *WT1*” in panel (i) and “Root → *NPM1* → *IDH2* → *PTPN11*” in

panel (iii) of Supplementary Figure S18 are in ties with many other events receiving the lowest probability. In the original implementation, we assign the best ranking to the tied events, but this approach can be too optimistic. For example, if there are three events with probabilities (0.01, 0.01, 0.98), then the percentile ranks would be (0.67, 0.67, 1) when all tied events receive the maximal ranking. On the contrary, if all tied events receive the minimal ranking, then the percentile ranks would be (0.33, 0.33, 1). Another definition would be to take the average ranking of the tied events, resulting in (0.5, 0.5, 1). While all three definitions are valid, we find the one with average ranking most representative of the predictive performance. We now clarify this in Methods:

p.14 *“... In the case of ties, we take the average percentile rank of the tied events.”*

After adjustment, we see that the standard deviation of the REVOLVER predictions is relatively large in both retrospective and forward predictions (Figure 6b & 6c) because there are many ties. The performance gain of TreeMHN over REVOLVER is more evident.

7. Regarding the github repo:

- a. Installation on Mac OS was not entirely smooth. There may be some path setting issues. We had to set `"/opt/homebrew/bin/g++-12"` to make the installation work. In general it would usually be `"/opt/homebrew/bin/g++-[whatever version]"`. It would be helpful to include this in Q&A or the guide.

We changed the instruction from `"CXX=g++-11"` to `"CXX=$(brew --prefix)/bin/g++-[INSTALLED VERSION]"`.

- b. The tutorial is detailed and runs smoothly. It reproduces a good portion of the results in the manuscript. Code had helpful inline comments with some computationally intensive steps accelerated using C++.

We would like to thank the reviewer for taking the time to verify the code.

- c. Some steps take a significant amount of time (~20 min). It would be helpful to mention the ballpark range of expected running time to give users a reasonable expectation.

We added the expected running times for the relevant code chunks.

- d. I encourage the authors to further develop the github front page so it can potentially attract more users.

We further developed the GitHub frontpage, including adding a new logo for TreeMHN and a quick start section.

Reviewer comments, second round

Reviewer #2 (Remarks to the Author):

My comments have been addressed. I believe the revised manuscript to be of strong interest to the readership of this journal.

Reviewer #3 (Remarks to the Author):

The authors have largely addressed our criticisms. Some very minor presentation issue, e.g., it will be helpful to also add "ancestor/descendant" to the matrix form in Fig. 1. In the Introduction, I am not sure trees from multi-regional sequencing of the same tumor can be regarded as independent trees. But I understand these are likely semantic difference and addressing longitudinal sampling issue is out of the scope of the work.

Reviewer #4 (Remarks to the Author):

I thank the Authors for revising and improving the manuscript, as well as performing additional experiments. I particularly appreciate that they have clearly marked (in red-blue text) what parts of the manuscript were edited, which facilitated reviewing on my end.

I am now satisfied with the experimental side of the work and have no additional major criticism, except some minor suggestions provided below.

Unfortunately, I am still unable to fully follow the methods description.

To start with a positive side, Page 12 and part of Page 13 are now clear, including Equations (1) - (3). I have some questions about Equation (1), but they are not related to the concerns described in the next paragraphs of this review. Algorithm presented in Supplementary Section A.1. is also clear and it is noticeable that the Authors put an effort in making sure that their pseudocode is correct and well documented.

However, when I move to Supplementary Section A.2. I get stuck. Already in Supplementary Equation (1) I can not properly link λ with Figure S1. Why is there (0,1,3) in some subscripts and (0,3) in others? When looking into $\lambda_{0,3}$ (fifth row, second column), I thought that (0,3) might be representing the edge present in the row tree and absent in the column tree. But then I checked the fourth row, second column entry and noticed that this reasoning is not correct. Considering that there is no precise definition of what $tree_2 \setminus tree_1$ is in Supplementary Equation (1) I am already feeling uncomfortable about proceeding. Then I come to Supplementary Equation (2), which I assume originates from some theoretical results on Markov chains but no details about its derivation nor easy-to-check reference are provided. There is a mention of unit vector e_{tree} with 1 at position of $tree$. What are the other positions in this vector corresponding to (if it is to some trees, which trees?) and what is its dimension? Then in Supplementary Equation (3), $p(tree | \Theta)$ is computed by using some formula given in the first line of the equation without any clear connection made between this formula and Equation (2) from the main text. Why is the probability given in Equation (2) in the main text equal to the formula given in the first line of Supplementary Equation (3)? Even change between line 2 and 3 in Supplementary Equation (3) is not well known. Where does this formula come from? While for this formula only a reference can be provided as it is a problem of integral calculation, thorough description and clear links should be established between the first line of Supplementary Equation (3) and Equation (2) in the main text. It is worth reiterating that I understand Equation (2) in the main text.

The main contributions of this paper are methodological and I believe that the Methods could have

a potential to attract attention of others and get adopted and extended in the future method developments. One of the main prerequisites for this is that the methods description is easy to follow and understand by a broader audience of Nature Communications, in particular method developers. I do not believe this is currently the case.

If the examples I provided above suggest that I am possibly lacking familiarity with some very basic mathematical concepts, I recommend the Authors to communicate with the Editorial Board on this matter and discuss potential addition of another reviewer or even excluding me from the rest of the review process of this work.

Importantly, I do not suspect that the Authors are hiding something or doing anything wrong. I am not hunting for mistakes, but I am just interested in understanding the entire methodological side of the work, which I am unable to do in its currently presented form.

I have a few additional comments, mostly minor. Two major comments are listed below.

(i) Can some motivation be provided for Equation (1)? Why do these variables follow the given distributions?

(ii) In Equation (1), for a given clone π_i , the emergence of clone (π_i, i) seems to be only dependent on the set of mutations defining π_i . But how biologically reasonable this assumption is? Isn't the overall subclonal composition of a tumor playing very important role? For example, if in tumor 1 subclone π_i has much higher growth dynamics/potential than other subclones, waiting time for (π_i, i) might be much shorter than in a tumor 2 in which π_i is growing slow compared to the other subclones.

MINOR

The sentence "For example, two parallel studies of acute myeloid leukemia (AML) using single-cell panel sequencing [20, 21] show that the reconstructed trees typically contain a small number of clones and can vary considerably between any two patients." can give an impression that AML samples typically contain a small number of clones. While this might be true, it can not be concluded based on the targeted single-cell data generated in the cited studies. So please either edit this sentence or provide a more relevant reference (if it exists). Analogous applies to the sentence starting on Page 5 and ending on Page 6, which states that trees reported by SCITE represent the complete evolutionary history of these tumors.

While I am familiar from REVOLVER from before, I think that a reader of this work not familiar with REVOLVER could find it difficult to get an idea about the main assumptions and goals of REVOLVER based on the brief summary of this method provided in the Introduction. I recommend rewriting this in order to make the summary of REVOLVER more clear and informative.

Please provide some basic information about the number of patients whose tumor development supports the following: "This observation aligns with previous studies, where some patients developed secondary resistance to FLT3 inhibitors due to off-target mutations (e.g. genes in the RAS pathways), which were present in small cell populations prior to treatment [37, 38]". In addition, what are the total sizes of the patient cohorts analyzed in the referenced studies? How many of the patients have mutations in the genes of interest?

I did not understand the purpose of some sentences, usually those related to the biological side. For example, why is the sentence "EGFR-mutant tumors with co-occurring TP53 were found to have higher degrees of genomic instability and shorter progression-free survival after EGFR TKI therapy [47]." relevant in the context where it is mentioned? Please revisit all sentences in the manuscript related to the biological side and keep only those that are informative and related to

this work. Also, make it more clear in what way TreeMHN relates to the mentioned biological observations and results.

I recommend adding brackets in the right hand side of Equation (3) so that it looks like $\text{argmax}_{\Theta} [\text{sum}_1 - \lambda \text{sum}_2]$.

Methods section, paragraph 2: "... evolutionary trajectories in π ..." -> "... evolutionary trajectory in π ..."

Authors' Response to Reviews of

Joint inference of exclusivity patterns and recurrent trajectories from tumor mutation trees

Xiang Ge Luo^{1,2}, Jack Kuipers^{1,2}, Niko Beerenwinkel^{1,2} (✉)

Manuscript #NCOMMS-22-28062A

doi.org/10.1101/2021.11.04.467347

Reviewer #2

My comments have been addressed. I believe the revised manuscript to be of strong interest to the readership of this journal.

We would like to thank the reviewer again for the time and effort in reviewing the manuscript and for the positive feedback.

Reviewer #3

The authors have largely addressed our criticisms. Some very minor presentation issue, e.g., it will be helpful to also add "ancestor/descendant" to the matrix form in Fig. 1. In the Introduction, I am not sure trees from multi-regional sequencing of the same tumor can be regarded as independent trees. But I understand these are likely semantic difference and addressing longitudinal sampling issue is out of the scope of the work.

We sincerely appreciate all constructive comments from the reviewer, which have helped us considerably improve the quality and the clarity of the manuscript.

As suggested by the reviewer, we have added "ancestor/descendant" to the matrix form in Fig. 1. In the Introduction, we meant to highlight the dependencies from multi-region sequencing samples of the same tumor encoded in one tree, which is not accounted for in previous CPMs but TreeMHN does, as opposed to one tree per sampling region. To make this clearer we referenced these dependencies at the end of the third and start of the last paragraphs of the Introduction.

Reviewer #4

I thank the Authors for revising and improving the manuscript, as well as performing additional experiments. I particularly appreciate that they have clearly marked (in red-blue text) what parts of the manuscript were edited, which facilitated reviewing on my end.

I am now satisfied with the experimental side of the work and have no additional major criticism, except some minor suggestions provided below.

We would like to thank the reviewer again for their suggestions of the additional experiments in improving the manuscript. We hope that the following response addresses all additional comments.

Unfortunately, I am still unable to fully follow the methods description.

To start with a positive side, Page 12 and part of Page 13 are now clear, including Equations (1) - (3). I have some questions about Equation (1), but they are not related to the concerns described in the next paragraphs

of this review. Algorithm presented in Supplementary Section A.1. is also clear and it is noticeable that the Authors put an effort in making sure that their pseudocode is correct and well documented.

We thank the reviewer for the positive feedback.

However, when I move to Supplementary Section A.2. I get stuck.

We have re-written supplementary section A.2 and explain the new structure before answering each specific question below. We had previously implicitly used the equivalence between the tree-generating process described in the main text and the continuous-time Markov chain with an independent exponentially-distributed sampling time defined in Supplementary Section A. However, we agree that we had not made this link explicit, which could make it harder for the readers to follow. We have now completely revised this and make the connection fully explicit. In particular we now split the original Supplementary Section A.2 into two sections. In the new Supplementary Section A.2, we provide precise definitions and notations associated with the continuous-time Markov chain and elaborate on its connection with the tree-generating process using Markov chain theory from the classical textbook by Norris [1998]. The new Theorem 1 establishes the equivalence between the definition of the Markov chain and the waiting time distributions of subclones given in main Equation (1). Specifically:

Supp. p.4 “

Theorem 1. *A right-continuous process $(X_t)_{t \geq 0}$ with values in the space of tumor mutation trees $S_{\mathcal{T}}^n$ is continuous-time Markov($\delta_{\mathcal{T}_0}, Q$) if and only if the waiting times of subclones $\pi \in \mathcal{T}$ for any $\mathcal{T} \in S_{\mathcal{T}}^n$ are exponential random variables such that*

$$T_{(0)} = 0, \quad T_{\pi} \sim T_{\text{pa}(\pi)} + \text{Exp}(\lambda_{\pi}). \quad (5)$$

”

Theorem 2 provides a formula for computing the marginal likelihood of a tree in main Equation (2):

Supp. p.7 “

Theorem 2. *Let $\Theta \in \mathbb{R}^{n \times n}$ be a Mutual Hazard Network, where $n \in \mathbb{N}^+$ is the number of mutations. Let $(X_t)_{t \geq 0}$ be continuous-time Markov($\delta_{\mathcal{T}_0}, Q$). Suppose $\mathcal{T} \in S_{\mathcal{T}}^n$ is an output of the tree generating process. Then the marginal probability of observing \mathcal{T} is given by*

$$p(\mathcal{T} \mid \Theta) := P \left(\max_{\pi \in \mathcal{T}} T_{\pi} < T_s < \min_{\pi' \in \text{Exit}(\mathcal{T})} T_{\pi'} \mid \Theta \right) = (\lambda_s (\lambda_s I - Q)^{-1})_{\mathcal{T}_0, \mathcal{T}}. \quad (6)$$

”

In the new Supplementary Section A.3, we further simplify the calculation of the likelihood using techniques from linear algebra.

All page and reference numbers in our response are based on the revised manuscript, unless otherwise stated. The page and reference numbers mentioned in the reviewers' comments are kept intact and refer to the original manuscript.

Already in Supplementary Equation (1) I can not properly link lambda with Figure S1. Why is there (0,1,3) in some subscripts and (0,3) in others? When looking into lambda_0,3 (fifth row, second column), I thought that (0,3) might be representing the edge present in the row tree and absent in the column tree. But then I checked the fourth row, second column entry and noticed that this reasoning is not correct. Considering that there is no precise definition of what tree_2 \setminus tree_1 is in Supplementary Equation (1) I am already feeling uncomfortable about proceeding.

This question concerns the definition and the interpretation of the generator matrix Q of the continuous-time Markov chain. In the revision, we first clarify that subclones in tumor mutation trees are sequences of mutations, and tumor mutation trees are sets of subclones. For example, the tree displayed in Supplementary Figure 1(b) is a set of four subclones $\{(0), (0, 1), (0, 3), (0, 1, 3)\}$. In particular, we use the following notation:

Supp. p.4 “ ...

- $\pi \in \mathcal{T}$ means that π is a subclone of the tree \mathcal{T} .
- We use $|\mathcal{T}|$ to denote the number of subclones in tree \mathcal{T} .
- We say that \mathcal{T}_1 is a subtree of \mathcal{T}_2 if all subclones in \mathcal{T}_1 can be found in \mathcal{T}_2 and write $\mathcal{T}_1 \subseteq \mathcal{T}_2$. If \mathcal{T}_1 is a subtree of \mathcal{T}_2 and $|\mathcal{T}_1| < |\mathcal{T}_2|$, then we write $\mathcal{T}_1 \subset \mathcal{T}_2$.
- We use $\mathcal{T}_2 \setminus \mathcal{T}_1$ to denote the set of subclones in \mathcal{T}_2 that are not subclones of \mathcal{T}_1 . The number of such subclones are denoted by $|\mathcal{T}_2 \setminus \mathcal{T}_1|$.

... ”

Then, we follow the terminology of Norris [1998] and define tumor progression as a continuous-time Markov chain $(X_t)_{t \geq 0}$ on the state space of all tumor mutation trees. In the new Supplementary Equation (2), the generator matrix Q is indexed by trees (the states) and defined as

$$q_{\mathcal{T}_1, \mathcal{T}_2} := \begin{cases} \lambda_{\mathcal{T}_2 \setminus \mathcal{T}_1} & \mathcal{T}_1 \subset \mathcal{T}_2, |\mathcal{T}_2 \setminus \mathcal{T}_1| = 1 \\ -\sum_{\mathcal{T} \neq \mathcal{T}_1} q_{\mathcal{T}_1, \mathcal{T}} & \mathcal{T}_1 = \mathcal{T}_2 \\ 0 & \text{otherwise.} \end{cases}$$

The direction of the transitions in the original Supplementary Equation (1) was from column states to row states. However, to be consistent with the definition of the Q -matrix in [Norris, 1998], the direction in the new Supplementary Equation (2) is from row states to column states, which is why the new Supplementary Figure S1(c) is an upper-triangular matrix, whereas the old one was lower-triangular. This direction now matches the reviewer’s intuition in their comment. The interpretation of the off-diagonal entries in Q is that:

Supp. p.4 “ ... transitioning from \mathcal{T}_1 to \mathcal{T}_2 is only possible if \mathcal{T}_2 has exactly one extra subclone than \mathcal{T}_1 , and this new subclone $\pi = \mathcal{T}_2 \setminus \mathcal{T}_1$ must be a child of one of the existing subclones in \mathcal{T}_1 ... ”

The transition rate $\lambda_{\mathcal{T}_2 \setminus \mathcal{T}_1}$ is not only associated with the mutation that labels the extra vertex, *i.e.* the most recent mutation in the subclone $\pi = \mathcal{T}_2 \setminus \mathcal{T}_1$, but actually all the mutations accumulated in the subclone, which was previously defined in Equation (1) of the main text and now reiterated in the new Supplementary Equation (3):

Supp. p.4 - 5 “ ... The transition rate is associated with all mutations in $\pi = (\text{pa}(\pi), i)$ for some $i \in [n] \setminus \text{pa}(\pi)$ and parameterized by a Mutual Hazard Network $\Theta \in \mathbb{R}^{n \times n}$,

$$\lambda_{\mathcal{T}_2 \setminus \mathcal{T}_1} = \lambda_\pi := \exp \left(\theta_{ii} + \sum_{j \in \text{pa}(\pi)} \theta_{ij} \right) = \Theta_{ii} \prod_{j \in \text{pa}(\pi)} \Theta_{ij}, \quad (3)$$

which is equivalent to the definition in Equation (1) of the main text (notation-wise shifted by one mutation).”

For example, the entry $q_{\mathcal{T}_1, \mathcal{T}_2}$ in the matrix shown in Figure 1 below is $\lambda_{(0,1,3)}$, which is different from the entry $q_{\mathcal{T}_1, \mathcal{T}_3} = \lambda_{(0,3)}$, even though the extra vertex in both \mathcal{T}_2 and \mathcal{T}_3 is the same mutation 3. This is because by our definition $\mathcal{T}_2 \setminus \mathcal{T}_1 = (0, 1, 3)$ but $\mathcal{T}_3 \setminus \mathcal{T}_1 = (0, 3)$.

Figure 1: The sub-matrix $Q_{\mathcal{T}}$, which corresponds to the tree \mathcal{T} in Supplementary Figure 1(b) and is defined in Supplementary Equation (36).

We would also like to emphasize that $Q_{\mathcal{T}}$ is only a sub-matrix of Q , where the columns and rows correspond to all subtrees of \mathcal{T} . The dimension of $Q_{\mathcal{T}}$ is much smaller than the dimension of Q , allowing us to minimize the dimension of matrix inversion for calculating the likelihood by replacing Supplementary Equation (6) with Supplementary Equation (13).

Then I come to Supplementary Equation (2), which I assume originates from some theoretical results on Markov chains but no details about its derivation nor easy-to-check reference are provided. There is a mention of unit vector $e_{\{\text{tree}\}}$ with 1 at position of $\{\text{tree}\}$. What are the other positions in this vector corresponding to (if it is to some trees, which trees?) and what is its dimension?

The reviewer is correct that the original Supplementary Equation (2), which is now the new Supplementary Equation (8), originates from Markov chain theory. This is now clarified in the proof of Theorem 2 with reference to [Norris, 1998]:

Supp. p.7 “Next, the matrix $P(t) = e^{tQ}$ is the transition probability matrix of $(X_t)_{t \geq 0}$, which by Theorem 2.1.1 of [Norris, 1998] solves the backward equation,

$$\frac{d}{dt}P(t) = QP(t), \quad P(0) = I. \quad (7)$$

The transition probability from \mathcal{T}_1 to \mathcal{T}_2 in time t is given by

$$P(X_t = \mathcal{T}_2 \mid X_0 = \mathcal{T}_1) = p_{\mathcal{T}_1, \mathcal{T}_2}(t),$$

where $p_{\mathcal{T}_1, \mathcal{T}_2}(t)$ is the $(\mathcal{T}_1, \mathcal{T}_2)$ entry in e^{tQ} , the row is indexed by \mathcal{T}_1 , and the column is indexed by \mathcal{T}_2 . In particular, starting with $X_0 = \mathcal{T}_0$, the probability of observing \mathcal{T} at sampling time t_s is

$$P(X_{t_s} = \mathcal{T} \mid X_0 = \mathcal{T}_0) = p_{\mathcal{T}_0, \mathcal{T}}(t_s) = (e^{t_s Q})_{\mathcal{T}_0, \mathcal{T}}, \quad (8)$$

which is the $(\mathcal{T}_0, \mathcal{T})$ entry in $e^{t_s Q}$.”

Like the generator matrix Q , the unit vector $e_{\mathcal{T}}$ is also indexed by the states of the Markov chain, which are the trees. The number of rows and columns in Q as well as the number of entries in $e_{\mathcal{T}}$ is size of the state space, which grows super-exponentially in the number of mutations n (see Supplementary Equation (1)). We introduce the unit vector $e_{\mathcal{T}}$ with 1 at the position of the tree \mathcal{T} and 0 at all other positions (one-hot encoding) in order to extract the $(\mathcal{T}_0, \mathcal{T})$ entry in the matrix $e^{t_s Q}$ via $e_{\mathcal{T}_0}^\top (e^{t_s Q}) e_{\mathcal{T}}$. We give an example of $e_{\mathcal{T}}$ in Figure 2 below, where the state space contains 9 trees for $n = 2$ mutations.

Figure 2: Example of a unit vector on the state space of tumor mutation trees for $n = 2$.

Then in Supplementary Equation (3), $p(\text{tree} \mid \Theta)$ is computed by using some formula given in the first line of the equation without any clear connection made between this formula and Equation (2) from the main text. Why is the probability given in Equation (2) in the main text equal to the formula given in the first line of Supplementary Equation (3)? Even change between line 2 and 3 in Supplementary Equation (3) is not well known. Where does this formula come from? While for this formula only a reference can be provided as it is a problem of integral calculation, thorough description and clear links should be established between the first line of Supplementary Equation (3) and Equation (2) in the main text. It is worth reiterating that I understand Equation (2) in the main text.

We agree that the equivalence between main Equation (2) and the original Supplementary Equation (3) for computing the marginal probability $p(\mathcal{T} \mid \Theta)$ was unclear. We now prove this result in Theorem 2:

Supp. p.7 “

Theorem 2. Let $\Theta \in \mathbb{R}^{n \times n}$ be a Mutual Hazard Network, where $n \in \mathbb{N}^+$ is the number of mutations. Let $(X_t)_{t \geq 0}$ be continuous-time Markov($\delta_{\mathcal{T}_0}, Q$). Suppose $\mathcal{T} \in S_{\mathcal{T}}^n$ is an output of the tree generating process. Then the marginal probability of observing \mathcal{T} is given by

$$p(\mathcal{T} \mid \Theta) := P \left(\max_{\pi \in \mathcal{T}} T_{\pi} < T_s < \min_{\pi' \in \text{Exit}(\mathcal{T})} T_{\pi'} \mid \Theta \right) = (\lambda_s (\lambda_s I - Q)^{-1})_{\mathcal{T}_0, \mathcal{T}}. \quad (6)$$

”

Here $\pi' \in \text{Exit}(\mathcal{T})$ implies $\pi' \notin \mathcal{T}$ and $\pi' \in A(\mathcal{T})$ by definition:

Supp. p.4 “Given a tree \mathcal{T} , we define the exit set of \mathcal{T} as

$$\text{Exit}(\mathcal{T}) = \{\pi \mid \pi \notin \mathcal{T}, \text{pa}(\pi) \in \mathcal{T}\},$$

which contains the children of the existing subclones that are not yet in \mathcal{T} , i.e. all subclones that could appear next. Then, we define the augmented tree as

$$A(\mathcal{T}) := \mathcal{T} \cup \text{Exit}(\mathcal{T}), \quad \text{i.e., } \text{Exit}(\mathcal{T}) = A(\mathcal{T}) \setminus \mathcal{T}.$$

”

The change from line 2 to 4 in the original Supplementary Equation (3) follows from the Laplace transform [Spiegel, 1965] of the backward equation $dP(t)/dt = QP(t)$ with $P(0) = I$ and the non-singularity of the matrix $\lambda_s I - Q$, which we now show in the proof of Theorem 2:

Supp. p.8 “Here, $\mathcal{L}(P(t)) = \int_0^\infty e^{-\lambda_s t_s} P(t_s) dt_s = \int_0^\infty e^{-\lambda_s t_s} e^{t_s Q} dt_s = A(\lambda_s)$ is the Laplace transform of $P(t)$. [...] Therefore, the marginal probability in main Equation (2) is

$$\begin{aligned} p(\mathcal{T} \mid \Theta) &= P \left(\max_{\pi \in \mathcal{T}} T_\pi < T_s < \min_{\pi' \in \text{Exit}(\mathcal{T})} T_{\pi'} \mid \Theta \right) \\ &= (\lambda_s A(\lambda_s))_{\mathcal{T}_0, \mathcal{T}} \\ &= (\lambda_s (\lambda_s I - Q)^{-1})_{\mathcal{T}_0, \mathcal{T}}. \end{aligned}$$

”

The main contributions of this paper are methodological and I believe that the Methods could have a potential to attract attention of others and get adopted and extended in the future method developments. One of the main prerequisites for this is that the methods description is easy to follow and understand by a broader audience of Nature Communications, in particular method developers. I do not believe this is currently the case.

If the examples I provided above suggest that I am possibly lacking familiarity with some very basic mathematical concepts, I recommend the Authors to communicate with the Editorial Board on this matter and discuss potential addition of another reviewer or even excluding me from the rest of the review process of this work.

Importantly, I do not suspect that the Authors are hiding something or doing anything wrong. I am not hunting for mistakes, but I am just interested in understanding the entire methodological side of the work, which I am unable to do in its currently presented form.

We greatly appreciate the reviewers’ questions and comments regarding Supplementary Section A and its connection with the main Methods section, allowing us to improve the clarity of the manuscript. The main purpose of introducing the Markov chain version is to provide a way to compute the likelihood. Considering the length of the manuscript, we decided to keep the tree-generating process as is in the main text, which facilitates readability for the general audience, and leave the supplement to the interested method developers. In the Methods section, we have added cross-references to the new content:

p.12 “Both $p(\mathcal{T} \mid \Theta)$ and its gradients $\partial p(\mathcal{T} \mid \Theta) / \partial \lambda_\pi$ can be computed efficiently by inverting specific triangular matrices, which are constructed using the rates associated with the events in $A(\mathcal{T})$, and the dimensions depend only on the number of subtrees in \mathcal{T} (see Theorem 1 and 2 in Supplementary Section A.2 & the derivations in Supplementary Section A.3).”

I have a few additional comments, mostly minor. Two major comments are listed below.

(i) Can some motivation be provided for Equation (1)? Why do these variables follow the given distributions?

In main Equation (1), we model the waiting time until a new mutation i occurs and fixates in an existing subclone π , resulting in a new subclone (π, i) , with exponential distributions. This is because the exponential has the memoryless property, meaning that the waiting time until an event happens does not depend on how much time has elapsed already, which is the most reasonable assumption to make when we know little about the time dependency of the process and a common choice for cancer progression models [Beerenwinkel and Sullivant, 2009, Beerenwinkel et al., 2015, Cristea et al., 2017, Schill et al., 2020]. The parameterization of the rates with a Mutual Hazard Network follows from the observation that mutations do not occur completely at random but exhibit patterns of clonal co-occurrence and exclusivity, which has been noted in the Introduction:

p.1 *“In particular, both studies report over-represented pairs of co-occurring or clonally exclusive mutations in the subclones. [...] Mutations that co-occur more frequently in the same clonal lineage may indicate synergistic effects on cell proliferation and survival [4]. Clonally exclusive mutations, on the other hand, occur more frequently in different lineages. They may suggest either clonal cooperation, where cell populations harboring complementary sets of mutations collaborate to promote tumor growth [23], or synthetic lethality, meaning that acquiring both mutations on the same genome will significantly reduce the viability of the clone [24]. Since clonal interactions have a major impact on intratumor heterogeneity and the observed evolutionary trajectories, it would be natural to incorporate mutational interdependencies within and between subclones when modelling tumor progression.”*

(ii) In Equation (1), for a given clone π , the emergence of clone (π, i) seems to be only dependent on the set of mutations defining π . But how biologically reasonable this assumption is? Isn't the overall subclonal composition of a tumor playing very important role? For example, if in tumor 1 subclone π has much higher growth dynamics/potential than other subclones, waiting time for (π, i) might be much shorter than in a tumor 2 in which π is growing slow compared to the other subclones.

While the rate of a subclone (π, i) depends only on the set of mutations defining (π, i) , the future state of the tumor depends on the set of subclones that could emerge next, which is determined by the current composition of the tumor, *i.e.* the exit set of the tree

Supp. p.4 *“Given a tree \mathcal{T} , we define the exit set of \mathcal{T} as*

$$\text{Exit}(\mathcal{T}) = \{\pi \mid \pi \notin \mathcal{T}, \text{pa}(\pi) \in \mathcal{T}\},$$

*which is the children of the existing subclones that are not yet in \mathcal{T} , *i.e.* all subclones that could appear next. Then, we define the augmented tree as*

$$A(\mathcal{T}) := \mathcal{T} \cup \text{Exit}(\mathcal{T}) \quad \Leftrightarrow \quad \text{Exit}(\mathcal{T}) = A(\mathcal{T}) \setminus \mathcal{T}.$$

”

This result follows directly from competing exponentials and is reflected in main Equation (14) as well as Supplementary Equation (6) and (9):

p.14 *“By competing exponentials, we calculate the probability of an event π happens before all the other events in $A(\mathcal{T}) \setminus \mathcal{T}$ as*

$$p(\pi \mid \mathcal{T}, \Theta) = \frac{\lambda_\pi}{\sum_{\pi' \in A(\mathcal{T}) \setminus \mathcal{T}} \lambda_{\pi'}} \times \mathbb{1}\{\pi \in A(\mathcal{T}) \setminus \mathcal{T}\}, \quad (14)$$

where λ_π is the rate associated with event π (Eq. (1)).”

Supp. p.5 “The diagonal entries are chosen such that the row sums of Q are zero. We interpret the off-diagonal row sum

$$q_{\mathcal{T}_1} := \sum_{\mathcal{T} \neq \mathcal{T}_1} q_{\mathcal{T}_1, \mathcal{T}} = \sum_{\pi \in \text{Exit}(\mathcal{T}_1)} \lambda_{\pi}, \quad (6)$$

as the rate of leaving the state \mathcal{T}_1 , which is the sum of the rates of all events that could happen next. [...] Furthermore, we define the jump matrix $\Xi = (\xi_{\mathcal{T}_1, \mathcal{T}_2} : \mathcal{T}_1, \mathcal{T}_2 \in S_{\mathcal{T}}^n)$ of Q as

$$\xi_{\mathcal{T}_1, \mathcal{T}_2} := \begin{cases} q_{\mathcal{T}_1, \mathcal{T}_2} / q_{\mathcal{T}_1} & \mathcal{T}_1 \neq \mathcal{T}_2 \text{ and } q_{\mathcal{T}_1} \neq 0 \\ 0 & \mathcal{T}_1 \neq \mathcal{T}_2 \text{ and } q_{\mathcal{T}_1} = 0, \end{cases} \quad (9)$$

$$\xi_{\mathcal{T}_1, \mathcal{T}_1} := \begin{cases} 0 & q_{\mathcal{T}_1} \neq 0 \\ 1 & q_{\mathcal{T}_1} = 0. \end{cases}$$

By Theorem 2.8.2 of [Norris, 1998], the jump chain $(Y_k)_{k \geq 0}$ associated with $(X_t)_{t \geq 0}$ is a discrete-time Markov chain with initial distribution $\delta_{\mathcal{T}_0}$ and jump matrix Ξ , where the entry $\xi_{\mathcal{T}_1, \mathcal{T}_2}$ is the probability of jumping from \mathcal{T}_1 to \mathcal{T}_2 .”

For example, suppose $\text{Exit}(\mathcal{T}_1) = \{\pi_1, \pi_2\}$ and $\text{Exit}(\mathcal{T}_2) = \{\pi_1, \pi_3\}$ with rates $\lambda_{\pi_1} = 1$, $\lambda_{\pi_2} = 0.5$, and $\lambda_{\pi_3} = 1.2$. The probability of π_1 appearing next in \mathcal{T}_1 is $\frac{1}{1+0.5} \approx 66.67\%$, which is greater than $\frac{1}{1+1.2} \approx 45.45\%$, the probability of π_1 appearing next in \mathcal{T}_2 . This property allows us to model heterogeneous growth patterns in tumors even though TreeMHN does not explicitly model subclone sizes, which has been noted in Discussion:

p.10 “Nevertheless, TreeMHN does not take into account the subclone sizes, which can be viewed as consequences of clonal selection [64]. As noted by [65], inferring mutation rates and fitness values jointly may have promise but is challenging. On the one hand, larger subclone sizes can be attributed to both their earlier appearance or higher fitness [65]. On the other hand, the mutation rates in cancer progression models are rates of evolution, which implicitly involve subclone fitness. Thus, any attempts to modify TreeMHN to model clonal selection need to be taken with caution.”

MINOR

The sentence "For example, two parallel studies of acute myeloid leukemia (AML) using single-cell panel sequencing [20, 21] show that the reconstructed trees typically contain a small number of clones and can vary considerably between any two patients." can give an impression that AML samples typically contain a small number of clones. While this might be true, it can not be concluded based on the targeted single-cell data generated in the cited studies. So please either edit this sentence or provide a more relevant reference (if it exists). Analogous applies to the sentence starting on Page 5 and ending on Page 6, which states that trees reported by SCITE represent the complete evolutionary history of these tumors.

We agree with the reviewer that the two cited studies only focus on a number of driver mutations of interests, omitting unknown driver mutations as well as passenger mutations. Together with sampling bias and noise, the trees reconstructed by phylogenetic methods may not be the true complete evolutionary history of the tumors. Hence, we modify those sentences as follows:

p.1 “For example, two parallel studies of acute myeloid leukemia (AML) using single-cell panel sequencing [20, 21] show that the reconstructed trees typically ~~contain~~ contained a small number of clones based on the specific driver mutations that were part of the panel and can vary that they varied considerably between any two patients.”

p.5 *“We assume that the mutation trees reconstructed by SCITE [33], a single-cell phylogenetic method, represent the complete evolutionary histories of the tumors.”*

While I am familiar from REVOLVER from before, I think that a reader of this work not familiar with REVOLVER could find it difficult to get an idea about the main assumptions and goals of REVOLVER based on the brief summary of this method provided in the Introduction. I recommend rewriting this in order to make the summary of REVOLVER more clear and informative.

We would like to thank the reviewer for this suggestion. Given the length of the manuscript, however, we prefer to focus on the parts of REVOLVER that are relevant to this work and refer the interested readers to the cited paper. Hence, we only made slight modifications to the summary of REVOLVER:

p.2 *“Based on transfer learning, REVOLVER [7] ~~reconciles the heterogeneous phylogenetic trees of~~ infers the phylogenetic trees for a cohort of patients simultaneously and reconciles the heterogeneous trees using a matrix summarizing the frequencies of all pairwise ancestor-descendant relationships across tumors and outputs the trees having the smallest distance to the matrix. The entries in the normalized matrix are empirical estimates of the probability of one mutation being the ancestor of another mutation, which can be used to compute the probability of a possible evolutionary trajectory.”*

Please provide some basic information about the number of patients whose tumor development supports the following: "This observation aligns with previous studies, where some patients developed secondary resistance to FLT3 inhibitors due to off-target mutations (e.g. genes in the RAS pathways), which were present in small cell populations prior to treatment [37, 38]." In addition, what are the total sizes of the patient cohorts analyzed in the referenced studies? How many of the patients have mutations in the genes of interest?

As suggested by the reviewer, we have added relevant patient statistics from the cited studies to the quoted sentence:

p.6 *“This observation aligns with previous studies [Kennedy and Smith, 2020], where some patients (e.g. 3/11 in [Peretz et al., 2019] and 15/41 in [McMahon et al., 2019]) developed secondary resistance to FLT3 inhibitors due to off-target mutations (e.g. genes in the RAS pathways), which were present in small cell populations prior to treatment [McMahon et al., 2019, Kennedy and Smith, 2020].”*

Here, [Kennedy and Smith, 2020] is a review paper on *FLT3* mutations in AML, in which the “Off-Target Resistance in Parallel or Downstream Pathways” section provides references to recent studies that support our observation, including [Peretz et al., 2019] and [McMahon et al., 2019]. As these aspects are covered in much more detail in the original cited paper and our work is method-oriented, we found the current presentation (interpretation of the results + references) to be more concise.

I did not understand the purpose of some sentences, usually those related to the biological side. For example, why is the sentence "EGFR-mutant tumors with co-occurring TP53 were found to have higher degrees of genomic instability and shorter progression-free survival after EGFR TKI therapy [47]." relevant in the context where it is mentioned? Please revisit all sentences in the manuscript related to the biological side and keep only those that are informative and related to this work. Also, make it more clear in what way TreeMHN relates to the mentioned biological observations and results.

We would like to thank the reviewer for the question and suggestion. One of the key goals of TreeMHN is to infer the patterns of clonal exclusivity and co-occurrence by estimating the Mutual Hazard Network from a cohort of tumor mutation trees (see Figures 1, 4, & 7). In real data applications, we try to associate the estimated patterns with known biological results and observe that they are often linked to clinical outcome.

For example, in the section where we analyzed the NSCLC dataset [Jamal-Hanjani et al., 2017, Caravagna et al., 2018], we first observe the co-occurrence between *EGFR* and *TP53* in Figure 7. Then, the sentence quoted by the reviewer relates this co-occurrence to worse prognosis, suggesting that TreeMHN results may be of interest to clinicians. The remaining sentences related to the biological side are for the same purpose. To clarify the link between TreeMHN patterns and the biological results, we have made the following modifications in the manuscript:

p.2 “*Our estimated exclusivity patterns and most probable evolutionary trajectories not only confirm previous biological findings but also provide new insights into the interdependencies of mutations, which could be informative for clinical decisions.*”

p.8 - 9 “*... EGFR-mutant tumors with co-occurring TP53 were found to have higher degrees of genomic instability and shorter progression-free survival after EGFR TKI therapy [...] These observations indicate that the tumor progression processes leading to the observed co-occurrence of the genes may have direct clinical consequences.*”

p.9 - 10 “*The estimated network on all trees captures combined signals, which are mainly driven by the largest subgroups with HR+/HER2- status (Supplementary Figure S22). [...] Therefore, TreeMHN is capable of extracting key patterns related to clinical outcome from highly heterogeneous tumor mutation histories.*”

I recommend adding brackets in the right hand side of Equation (3) so that it looks like $\text{argmax}_{\Theta} [\text{sum}_1 - \lambda \text{sum}_2]$.

We have added the brackets to Equation (3).

Methods section, paragraph 2: "... evolutionary trajectories in π ..." -> "... evolutionary trajectory in π ..."

We have corrected this typo.

References

- N. Beerenwinkel and S. Sullivan. Markov models for accumulating mutations. *Biometrika*, 96:645–661, 2009.
- N. Beerenwinkel, R. F. Schwarz, M. Gerstung, and F. Markowetz. Cancer evolution: mathematical models and computational inference. *Systematic Biology*, 64:e1–e25, 2015.
- G. Caravagna, Y. Giarratano, D. Ramazzotti, I. Tomlinson, T. A. Graham, G. Sanguinetti, and A. Sottoriva. Detecting repeated cancer evolution from multi-region tumor sequencing data. *Nature Methods*, 15:707–714, 2018.
- S. Cristea, J. Kuipers, and N. Beerenwinkel. pathTiMEx: joint inference of mutually exclusive cancer pathways and their progression dynamics. *Journal of Computational Biology*, 24:603–615, 2017.
- M. Jamal-Hanjani, G. A. Wilson, N. McGranahan, N. J. Birkbak, T. B. Watkins, S. Veeriah, S. Shafi, D. H. Johnson, R. Mitter, R. Rosenthal, et al. Tracking the evolution of non-small-cell lung cancer. *New England Journal of Medicine*, 376:2109–2121, 2017.
- V. E. Kennedy and C. C. Smith. FLT3 mutations in acute myeloid leukemia: key concepts and emerging controversies. *Frontiers in Oncology*, 10:2927, 2020.
- C. M. McMahon, T. Ferng, J. Canaani, E. S. Wang, J. J. Morrisette, D. J. Eastburn, M. Pellegrino, R. Durruthy-Durruthy, C. D. Watt, S. Asthana, E. A. Lasater, R. DeFilippis, C. A. Peretz, L. H. McGary, S. Deihimi, A. C. Logan, S. M. Luger, N. P. Shah, M. Carroll, C. C. Smith, and A. E. Perl. Clonal selection with RAS pathway activation mediates secondary clinical resistance to selective FLT3 inhibition in acute myeloid leukemia. *Cancer Discovery*, 9:1050–1063, 2019.
- J. R. Norris. *Markov chains*. Cambridge university press, 1998.
- C. A. C. Peretz, L. H. McGary, T. F. Kumar, J. H. Jackson, J. Jacob, R. Durruthy-Durruthy, C. Zhang, M. J. Levis, A. E. Perl, A. Y. H. Leung, et al. Single cell sequencing reveals evolution of tumor heterogeneity of acute myeloid leukemia on quizartinib. *Blood*, 134:1440, 2019.
- R. Schill, S. Solbrig, T. Wettig, and R. Spang. Modelling cancer progression using Mutual Hazard Networks. *Bioinformatics*, 36:241–249, 2020.
- M. R. Spiegel. *Laplace transforms*. McGraw-Hill New York, 1965.

Reviewer comments, third round

Reviewer #4 (Remarks to the Author):

I thank the Authors for addressing all of my comments and wish them luck with publishing this work.

Authors' Response to Reviews of

Joint inference of exclusivity patterns and recurrent trajectories from tumor mutation trees

Xiang Ge Luo^{1,2}, Jack Kuipers^{1,2}, Niko Beerenwinkel^{1,2}(✉)

Manuscript #NCOMMS-22-28062B

doi.org/10.1101/2021.11.04.467347

Reviewer #4

I thank the Authors for addressing all of my comments and wish them luck with publishing this work.

We greatly appreciate the time and effort the reviewer spent in reviewing the manuscript, and we thank the reviewer for the positive feedback.